# Microbial ecology of northern Gulf of Mexico estuarine waters

**Michael W. Henson,[1] J. Cameron Thrash[2]**

**ABSTRACT**  Estuarine and coastal ecosystems are of high economic and ecological importance, owing to their diverse communities and the disproportionate role they play in carbon cycling, particularly in carbon sequestration. Organisms inhabiting these environments must overcome strong natural fluctuations in salinity, nutrients, and turbidity, as well as numerous climate change-induced disturbances such as land loss, sea level rise, and, in some locations, increasingly severe tropical cyclones that threaten to disrupt future ecosystem health. The northern Gulf of Mexico (nGoM) along the Louisiana coast contains dozens of estuaries, including the Mississippi–Atchafalaya River outflow, which dramatically influence the region due to their vast upstream watershed. Nevertheless, the microbiology of these estuaries and surrounding coastal environments has received little attention. To improve our understanding of microbial ecology in the understudied coastal nGoM, we conducted a 16S rRNA gene amplicon survey at eight sites and multiple time points along the Louisiana coast and one inland swamp spanning freshwater to high brackish salinities, totaling 47 duplicated Sterivex (0.2–2.7 μm) and prefilter (>2.7 μm) samples. We cataloged over 13,000 Amplicon Sequence ariants (ASVs) from common freshwater and marine clades such as SAR11 (Alphaproteobacteria), *Synechococcus* (Cyanobacteria), and acI and *Candidatus* Actinomarina (Actinobacteria). We observed correlations with freshwater or marine habitats in many organisms and characterized a group of taxa with specialized distributions across brackish water sites, supporting the hypothesis of an endogenous brackish-water community. Additionally, we observed brackish-water associations for several aquatic clades typically considered marine or freshwater taxa, such as SAR11 subclade II, SAR324, and the acI Actinobacteria. The data presented here expand the geographic coverage of microbial ecology in estuarine communities, help delineate the native and transitory members of these environments, and provide critical aquatic microbiological baseline data for coastal and estuarine sites in the nGoM.

**IMPORTANCE**  Estuarine and coastal waters are diverse ecosystems influenced by tidal fluxes, interconnected wetlands, and river outflows, which are of high economic and ecological importance. Microorganisms play a pivotal role in estuaries as "first responders" and ecosystem architects, yet despite their ecological importance, they remain underrepresented in microbial studies compared to open ocean environments. This leads to substantial knowledge gaps that are important for understanding global biogeochemical cycling and making decisions about conservation and management strategies in these environments. Our study makes key contributions to the microbial ecology of estuarine and coastal habitats in the northern Gulf of Mexico. Our microbial community data support the concept of a globally distributed, core brackish microbiome and emphasize previously underrecognized brackish-water taxa. Given the projected worsening of land loss, oil spills, and natural disasters in this region, our results will serve as important baseline data for researchers investigating the microbial communities found across estuaries.

Address correspondence to Michael W. Henson, mhenson@niu.edu, or J. Cameron Thrash, thrash@usc.edu.

The authors declare no conflict of interest.

See the funding table on p. 15.

**KEYWORDS**  bacterioplankton, microbial ecology, estuary microbiology, SAR11 clade, *Synechococcus*, brackish microbiome, mesohaline, brackish

Estuarine environments are highly diverse, interconnected ecosystems that are exposed to strong natural fluctuations in salinity and nutrient availability (1–3). Under future climate scenarios, estuaries are expected to have increased erosion and intrusion of saltwater from sea level rise, degraded water quality from runoff and pollution from severe storms, and increased temperatures (4–7), all of which could fundamentally alter microbial community composition and, thus, the processing of nutrients and food web dynamics (8). Fluctuations in salinity from sea level rise are especially likely to change the microbial community structure and metabolic capacity (8–11). This reflects the fact that organisms require unique metabolic and cellular features in marine and freshwater environments (6, 8, 12–15). These important differences are hypothesized to lead to low species diversity at intermediate salinities and high species diversity in freshwater and marine environments, as first visualized in the Remane curve (16). Although the Remane curve was not originally formulated to describe microorganisms, the pattern was found to apply to phytoplankton and other microbes (17). The relationships described in the curve anticipated the now-supported observation of infrequent transitions between marine and freshwater species (9, 15, 17, 18) and the importance of salinity in structuring microbial communities (8–11, 19).

Previous work has sought to investigate if brackish environments host autochthonous (native) microbial communities uniquely adapted to these fluctuating ecosystems. Indeed, brackish bacterial communities are genomically and taxonomically differentiated from their freshwater and marine counterparts (3, 6, 13, 19–27). For instance, taxa within the dominant clades *Synechococcus* and SAR11, specifically subcluster 5.2 and subclade III, have unique brackish ecotypes and corresponding genomic capacities such as pigment type, nitrogen utilization, and osmotic regulation that may have facilitated their transition into these dynamic environments (6, 21, 23, 25, 27). Moreover, because of the strong selection for factors influencing marine–freshwater transitions (9), genomic plasticity may be critical to these brackish-adapted taxa to maintain abundances and large biogeographic distribution at intermediary salinities (6). Therefore, estuarine microorganisms can be phylogenetically distinct and possess unique genomic capacities to adapt to natural fluctuations in environmental conditions (6, 25). However, the number of brackish ecosystems studied has been limited (6), necessitating the collection of more widespread baseline microbial community data to understand the impacts of future climate scenarios on the biogeographic distribution and functions of resident microorganisms.

Influenced by two major rivers—the Mississippi and Atchafalaya Rivers—and a vast network of interconnected wetlands, the northern Gulf of Mexico (nGoM) coastline is subject to continuous fluctuations in environmental conditions such as salinity, nutrients, and turbidity (28–30). The natural gradients created from these dynamic environments and their high economic and ecological importance make the northern Gulf of Mexico an excellent coastal/estuarine study system (31–34). Over the past century, nGoM wetlands have lost an estimated 5,000 km$^2$ of land due to erosion (35). Ecosystem models predicting future climate scenarios suggest continued loss of freshwater wetlands due to saltwater intrusion (4, 35) that will impact ecosystem functions across the coastal area, including biogeochemical cycling by microorganisms. However, previous microbiological research in this region has mostly focused on the communities associated with oil spills and eutrophication, and there is much less data on the general microbial ecology from the estuary ecosystems across the nGoM (36–41). Samples from pre-Deepwater Horizon oil spill microbial communities along the continental Louisiana shelf were dominated by Alphaproteobacteria and Bacteroidota, specifically the SAR11 clade (30). In contrast, deeper samples found an increasing abundance of Archaea, specifically *Crenarchaeota* (a.k.a. Thaumarchaeota and Nitrososphaerota) (42). However, limited investigation of the

microbial ecology from nGoM estuaries has restricted our ability to infer how bacterio-plankton in the nGoM naturally fluctuates over time in these diverse habitats.

To characterize the baseline microbial communities that inhabit the nGoM along the Louisiana coast and contribute more broadly to understanding estuarine and brackish water communities globally, we collected surface water at nine coastal, estuarine, and freshwater sites over multiple years and seasons. Sites sampled annually for 3 years included Lake Borgne (LKB; Pontchartrain watershed), Bay Pomme d'Or (JLB; Barataria watershed), Terrebonne Bay (TBON; Terrebonne watershed), Atchafalaya River Delta (ARD; Vermilion-Teche/Atchafalaya watershed), Freshwater City (FWC; Mermentau watershed), and Calcasieu Jetties (CJ; Calcasieu watershed), while sites Sabine Wetlands (Sabine; Calcasieu watershed) and Bay Batiste (BBAT; Barataria watershed) were sampled bi-monthly in 2015, and the inland Lake Martin (Swamp; Vermilion-Teche watershed) was sampled once in 2014. We assessed the particle-associated (>2.7 µm) and free-living (0.2–2.7 µm) fractions of the microbial communities as well as the associated water chemistry. Our results showcase the diversity of microbes inhabiting these environments that are adapted to different salinity regimes. While numerous clades show typical marine and freshwater associations, other taxa displayed a euryhaline distribution (found across a wide range of salinities) that supports the developing hypothesis of a core group of autochthonous taxa uniquely adapted to brackish environments (6, 8, 13, 25, 27). This study is a comprehensive description of the diverse microbial communities from nGoM estuarine and coastal systems and expands our biogeographic understanding of important and poorly characterized clades from these environments.

## RESULTS

We sampled nine sites [47 samples size-fractionated at 0.22–2.7 µm (24 samples) and >2.7 µm (23 samples)] from across southern Louisiana's coastal, estuarine, and

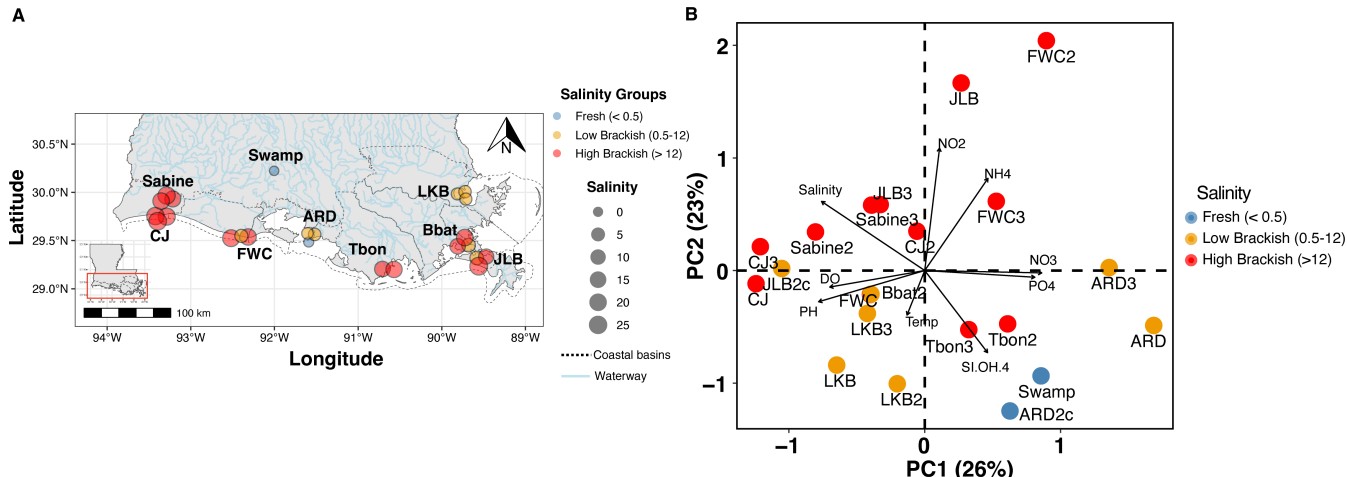

**FIG 1** (A) Locations of the nine sampling sites along the coast of the northern Gulf of Mexico. The shape of the point is the year the sample was collected. The color indicates the broad salinity classification of fresh (blue,<0.5 salinity), low brackish (orange,<12 salinity), and high brackish (red,>12 salinity) at the sampling site. The size of the shape corresponds to the measured salinity. The dotted line outlines the 10 coastal basins. The light blue lines are waterways. The inset is a map of Louisiana highlighting the targeted location of the sampling sites. The map was made with the R packages rnaturalearth and ggplot2. Shapefiles were obtained from U.S. Geological Survey, National Wetlands Research Center. The sites sampled were Lake Borgne (LKB, Shell Beach, LA) from the Pontchartrain watershed, Bay Pomme d'Or (JLB, Buras, LA) from the Barataria watershed, Terrebonne Bay (TBON, Cocodrie, LA) from the Terrebonne watershed, Atchafalaya River Delta (ARD, Franklin, LA) from the Vermilion-Teche/Atchafalaya watershed, Freshwater City (FWC, Kaplan, LA) from Mermentau watershed, Calcasieu Jetties (CJ, Cameron, LA) from the Calcasieu Watershed, Sabine Wetlands (Sabine, Cameron, LA) from the Calcasieu watershed, Bay Batiste (BBAT, Port Sulphur, LA) from the Barataria watershed, and Lake Martin (Swamp, Breaux Bridge, LA) from the Vermilion-Teche watershed (Table 1). (B) Two-dimensional principal coordinates analysis plot of normalized water characteristic variables measured at each site. Eigenvectors are scaled to strength. The percent variation of each principal coordinate explained is indicated in parentheses adjacent to the component axis. The color indicates the broad salinity classification of fresh (blue,<0.5 salinity), low brackish (orange,<12 salinity), and high brackish (red,>12 salinity) at the sampling site.

**TABLE 1** Site environmental measurements and location

| Site | Temperature range (°C) | Salinity range | Free-living fraction sequences | Particle-associated fraction sequences | Sampling dates (month/year) | Location (Lat., Long.) | Notes |
|---|---|---|---|---|---|---|---|
| ARD | 8.76–24.7 | 0.18–3.72 | Published[a] | Unpublished | 7/2015, 6/2016, 12/2016 | 29.575, −91.538 | Vermillion Bay, Atchafalaya River Delta (Burns Point, LA) |
| BBAT | 20.61–30.42 | 6.92–19.09 | Unpublished | Unpublished | 7/2015, 9/2015, 11/2015 | 29.458, −89.815 | Bay Batiste (Port Sulphur, LA) |
| CJ | 25.54–31.47 | 22.16–24.63 | Published[a] | Unpublished | 9/2014, 9/2015, 9/2016 | 29.760, −93.340 | Calcasieu Jetties (Cameron, LA) |
| FWC | 19.91–22.9 | 5.39–20.9 | Published[a] | Unpublished | 3/2015, 4/2016, 11/2016 | 29.530, −92.326 | Freshwater City, LA |
| JLB | 7.66–27.08 | 6.89–26.01 | Published[a] | Unpublished | 1/2015, 5/2015, 1/2017 | 29.348, −89.538 | Bay Pomme d'Or (Buras, LA) |
| LKB | 19.29–30.15 | 2.39–3.55 | Published[a] | Unpublished | 6/2015, 7/2016, 2/2017 | 30.003, −89.826 | Lake Borne, Shell Beach, LA |
| Sabine | 25.81–30.42 | 16.38–22.96 | Unpublished | Unpublished | 7/2015, 9/2015, 10/2015 | 29.920, −93.381 | Sabine National Wildlife Refuge (Huckberry, LA) |
| Swamp | 17.12 | 0 | Unpublished | Unpublished | 11/2014 | 30.221, −91.906 | Lake Martin (Beaux Bridge, LA) |
| TBON | 29.64–31.63 | 14.2–17.7 | Published[a] | Unpublished | 8/2015, 7/2016 | 29.207, −90.647 | Terrebonne Bay (Cocodrie, LA) |

[a](14, 43, 44).

swamp environments over 3 years from 2014 to 2017 (Fig. 1A; Table 1). Six sites were sampled once a year for 3 years as part of our nGoM cultivation campaign to compare our isolates to the natural communities, but a comprehensive ecological analysis of these samples was not previously completed (43, 44) (Table 1). Two other sites were sampled three times over 5 months, and one site, an inland swamp, was sampled only once (Table 1; Table S1). We found positive correlations between nitrate ($NO_3^-$) and phosphate ($PO_4^{2-}$) ($R = 0.649$, $P = 0.001$) and nitrite ($NO_2^-$) and ammonium ($NH_4^+$) ($R = 0.618$, $P = 0.003$) (Table S1: Nut_Cor), while temperature and dissolved oxygen ($R = -0.696$, $P < 0.005$) and salinity and silicic acid [$Si(OH)_4$] were negatively correlated ($R = -0.563$, $P < 0.001$) (Table S1: Nut_Cor). Sites ranged from fresh (<0.5 salinity) to high brackish (i.e., polyhaline) waters (max salinity = 26.01 at JLB) (Fig. 1A).

Principal component analysis of environmental conditions at the nine sites showed a distinct separation along both the PC1 and PC2 axes, which together explained more than half of the variance (Fig. 1B). Salinity and $NO_2^-$ were the most important variables separating the sites. Sites along the positive PC2 axis line typically had higher salinities (>12) and higher $NO_2^-$ and $NH_4^+$ concentrations. In contrast, sites along the negative PC2 axis were more indicative of high silicic acid (Fig. 1B). Sites along the positive PC1 axis typically had higher nitrate and phosphate concentrations (Fig. 1B).

After quality filtering and rarifying, we recovered 7,341 ASVs from the nine sites. ASVs were predominately classified into four bacterial phyla (82% of all reads)—Proteobacteria (36%), Actinobacteriota (20%), Bacteroidota (15%), and Cyanobacteria (11%)—and three archaeal phyla—Thermoplasmatota (0.3%), Crenarchaeota (0.3%), and Nanoarchaeota (0.2%). Non-metric multidimensional scaling (NMDS) of the 47 communities showed three distinct groupings based on salinity: fresh (<0.5 salinity), low brackish (0.5–12 salinity), and high brackish (>12 salinity) (Fig. 2; NMDS stress 0.135; ANOSIM $R = 0.703$, $P = 0.001$). Moreover, salinity ($R^2 = 0.807$, $P = 0.001$) and silicic acid ($R^2 = 0.563$, $P = 0.001$) were the two strongest environmental variables correlated to the NMDS ordination of the combined fraction analysis (Fig. 2; Table S1: envfit). While "brackish" is usually defined as salinities between 0.5 and 30, we observed a significant separation at the vertical axis of our NMDS ordination between sites with salinities above and below 12 and, thus, used this to separate high from low brackish categories. Filter fraction difference (free-living vs particle-associated) was significant but had low explanatory power (ANOSIM $R = 0.244$, $P = 0.001$).

## Free-living microbial communities

The free-living microbial communities examined separately from the particle-associated fractions were strongly differentiated by salinity ($R^2 = 0.842$, $P = 0.001$), silicic acid ($R^2 = 0.475$, $P = 0.016$), and temperature ($R^2 = 0.386$, $P = 0.005$) (Table S1: envfit). The majority of reads in the free-living fraction classified to the clades SAR11 (Alphaproteobacteria),

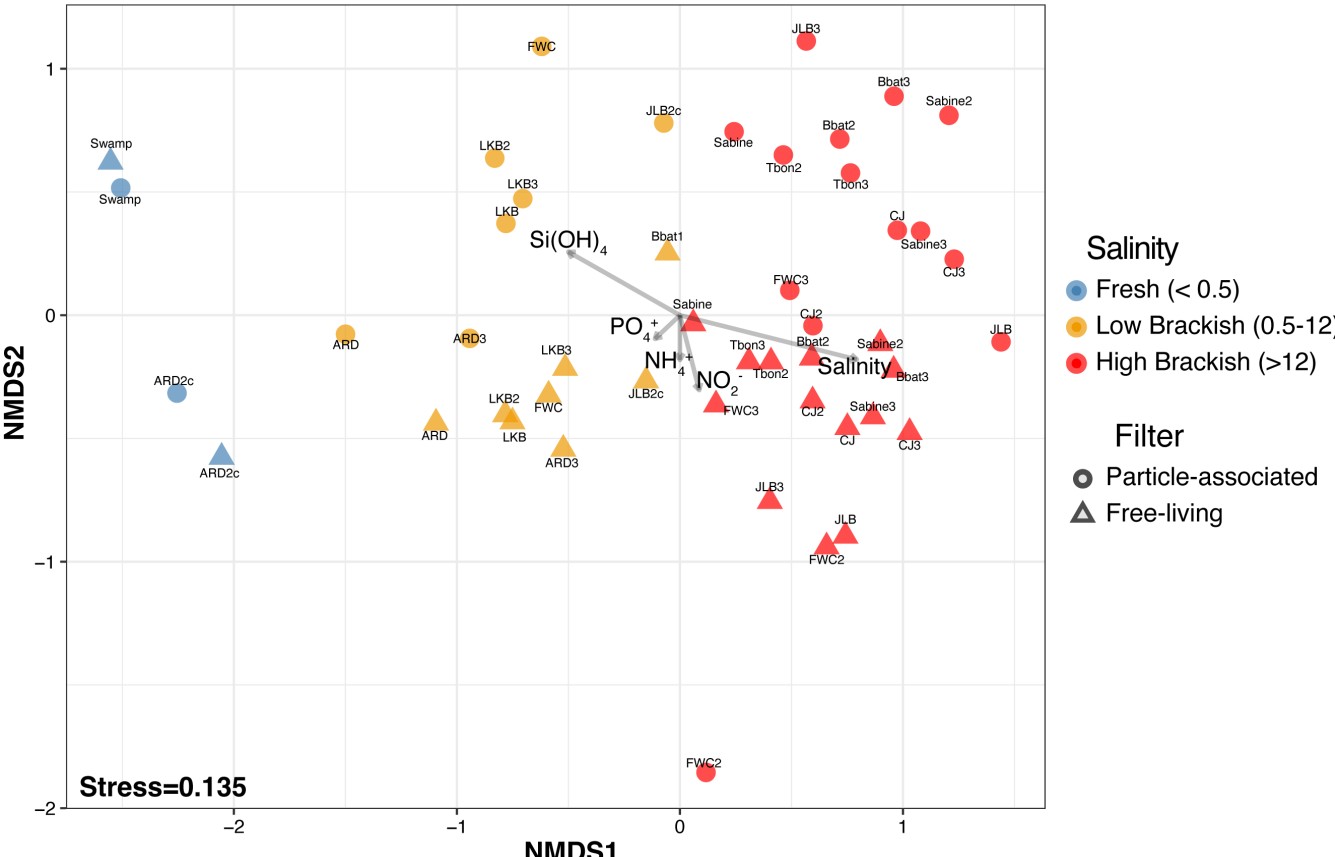

**FIG 2** Non-metric multidimensional scaling ordination of the microbial communities of the nine sites. The dot color indicates the broad salinity classification of fresh (blue,<0.5 salinity), low brackish (orange,<12 salinity), and high brackish (red,>12 salinity) at the sampling site. The dot shape indicates the community size fraction: triangle (free-living) and circle (particle-associated). Significant environmental variables (*P* < 0.05) determined with *envfit* are plotted as vectors. Arrow lengths have been adjusted based on their strength of correlation ($R^2$).

OM43 (Betaproteobacteria), *Candidatus* Actinomarina and acI (Actinobacteria), and *Cyanobium* (*Synechococcus* subcluster 5.2, Cyanobacteria) (Fig. 3A; Table S1: free-living RA). At higher salinities (>12), typical marine clades such as SAR11 subclade I (Alphaproteobacteria) and *Candidatus* Actinomarina (OM1 clade, Actinobacteria) were more abundant than in samples of lower salinity (Fig. 4A), supporting previous observations of their high relative abundance in samples from the Louisiana continental shelf with high salinity (30). In contrast, typical freshwater clades, such as SAR11 subclade IIIb/LD12 (*Candidatus* Fonsibacter sp., Alphaproteobacteria) and acI and acIV (Actinobacteria), were more abundant in lower salinities (Fig. 4A; Fig. S1; Table S1: free-living RA). Other ASVs with differentiated relative abundances were from the SAR86 (Gammaproteobacteria) and SAR324 clades, which both had higher relative abundance in higher salinity. ASV54 unknown *Holophagaceae* (Acidobacteria) and ASV13 *Candidatus* Aqualuna (Actinobacteria) were correlated with freshwater habitats (Fig. 4A; Table S1: free-living RA).

Taxa within the SAR11 and *Synechococcus* clades had diverse salinity habitat associations (Fig. 4A; Fig. S1). ASVs from SAR11 subclade I increased in relative abundance with increasing salinity, while we observed subclade IIIa ASVs distributed more broadly across all brackish environments (Fig. 4A and 5A; Fig. S1; Table S1: SAR11 RA; Table S1: pairwise). For instance, we observed ASV5 (SAR11 subclade IIIa.3) at relative abundances >1% at sites ranging from low (>0.5) to the maximum salinity sampled (26.01), overlapping with subclade I, while ASV31 (SAR11 subclade IIIa.3) occurred predominantly in lower salinity sites. We observed SAR11 subclade II-associated ASVs (e.g., ASV9 and ASV119) at high relative abundances across a breadth of salinities, complementing and expanding on previous descriptions of its distribution in aquatic habitats of varying salinities (27, 45–

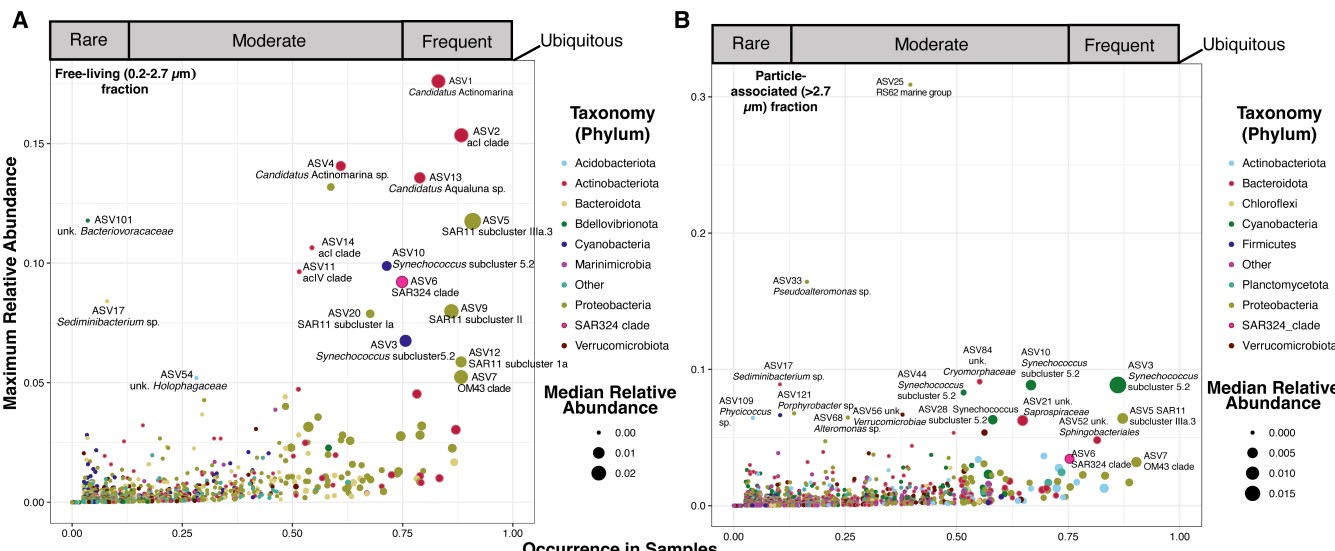

**FIG 3** Maximum relative abundance of each ASV in the free-living fraction (A) and particle-associated fraction (B) according to its percent occurrence (>0%) across samples and maximum relative abundance. ASVs are color-coded by phylum, and the size of the dot corresponds to the median relative abundance for each ASV.

48) (Fig. 5A; Fig. S1; Table S1: SAR11 RA). *Synechococcus* ASVs were predominately from subcluster 5.2, with only one ASV classified as subcluster 5.1 (belonging to the better-studied marine-specific group) (20, 21, 49, 50) (Fig. 5B; Table S1: *Synechococcus* RA). We observed the *Synechococcus* subcluster 5.2 ASV3 at >1% relative abundance at most of the sites sampled, whereas we found the ASV70 *Synechococcus* subcluster 5.1 only at low relative abundances in salinities >12. *Synechococcus* subcluster 5.2 ASV47 occurred in a limited number of samples (less than half) but a broad range of salinities (Fig. 5B).

## Particle-associated communities

Particle-associated communities were composed of bacteria classified as Cyanobacteria, Proteobacteria, Actinobacteria, and Bacteroidota (Fig. 3B; Table S1: >2.7 µm RA). We

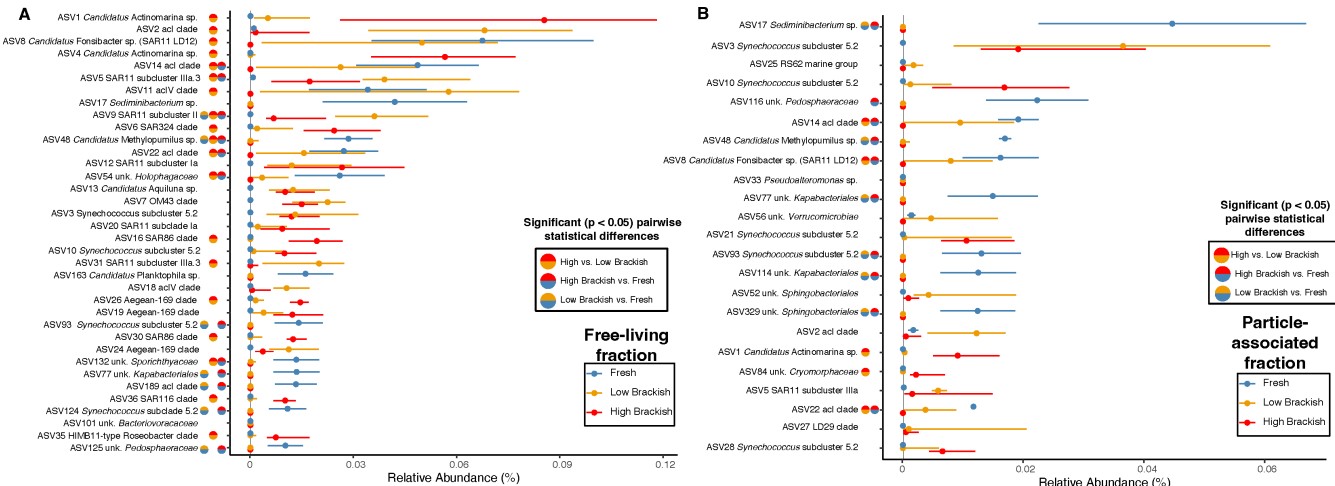

**FIG 4** ASVs (>1%RA) with significant differential relative abundance in fresh (blue,<0.5 salinity), low salinity (orange,<12 salinity), or high salinity (red,>12 salinity) in the free-living (A) and particle-associated (B) fractions according to the non-parametric, one-way ANOVA on ranks (Krustal–Wallis test). Bi-colored circles (red, orange = highvs low brackish; orange, blue = low brackish, fresh; red, blue = high brackish, fresh) indicate significant (P < 0.05) pairwise statistical differences between salinity groups controlling for the false discovery rate (Wilcoxon test with Benjamini–Hochberg correction) (Table S1). Points represent median values, and lines represent the interquartile range. The vertical line indicates the limit of detection.

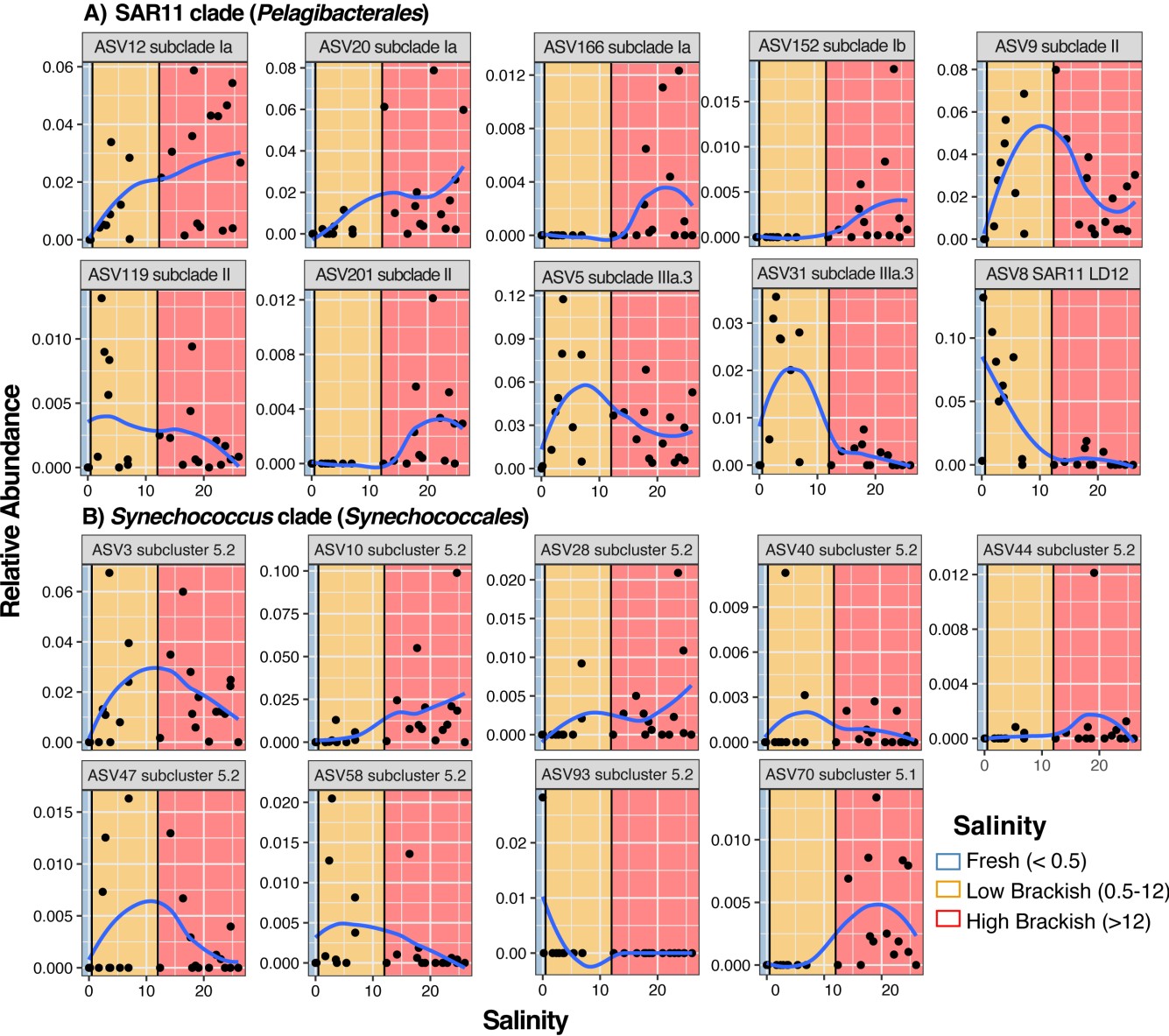

**FIG 5** Free-living fraction ASV relative abundance within key taxonomic clades: (A) SAR11 and (B) *Synechococcus* along the salinity spectrum sampled. Non-linear regression lines were generated using *geom_smooth* and the method *loess* within the *ggplot* function as a visual aid for relative abundance trends. ASVs are ordered by subclade (SAR11) or subcluster (*Synechococcus*) and then listed in numerical ASV order.

observed numerous taxa known for free-living lifestyles at high relative abundances in the particle-associated size fraction (e.g., *Cyanobium* and the SAR11 clade; Fig. S1) (2, 49, 51). Within the top 25 abundant taxa in the particle-associated fraction (average relative abundance), 13 were classified as known planktonic organisms, with median relative abundances >0.1% and max abundances between 0.25% and 3%, representing about 11% of the total reads (Fig. S1; Table S1: particle-associated RA). The presence of typical planktonic taxa within the particle-associated fraction complicated our interpretation of prefilter community diversity measurements. While organisms may have been present in this fraction due to cell size or their attachment to large particles, the presence of taxa such as the SAR11 and *Synechococcus* clades with known cell sizes of <2.7 µm and planktonic lifestyles (14, 27, 52–54) suggests that the high sediment loads found in the nGoM may have acted as an additional filter (Fig. 3B and 4B). Indeed, the amount of volume filtered can act as a secondary filter, trapping planktonic cells on the prefilter (>2.7 µm fraction) and biasing downstream analyses (55). While turbidity was not part of

the environmental parameters we collected, high turbidity is common across the nGoM, owing to the influence of runoff and the Mississippi and Atchafalaya Rivers (56, 57). Moreover, SAR11, *Synechococcus*, and SAR86—all known planktonic organisms—have been observed in particle-associated samples from the Amazon River plume, which may also represent a similar effect (58–60) Although we did not investigate the impact of filtered volume on which taxa were observed in the 2.7-μm filters, our results highlight that microbial ecologists should consider, and experimentally validate, how sediment load, in addition to volume, impacts size fractionated communities when working in coastal and estuary environments with high turbidity.

The particle-associated communities had a significantly higher species richness (Fig. S2) than the free-living fraction, which is typical of particle-associated communities (2, 61). However, the incorporation of free-living taxa into the larger filter fraction may have also increased particle-associated community richness. Statistically, the two size-fractionated communities were more strongly differentiated by salinity than filter fraction (salinity: ANOSIM $R = 0.82$, $P = 0.001$; filter: ANOSIM $R = 0.244$, $P = 0.001$), a result that may be partially explained by the presence of free-living taxa in the particle-associated fraction. When we examined the particle-associated communities alone, salinity and silicic acid were the most significant factors driving separation (salinity: $R^2 = 0.812$, $P = 0.001$; silicic acid: $R^2 = 0.686$, $P = 0.001$), with $NO_2^-$ ($R^2 = 0.482$, $P = 0.006$) and $NH_4^+$ ($R^2 = 0.354$, $P = 0.022$) also correlating to a lesser extent with the ordination (Fig. 2; Table S1: envfit).

Despite the numerous abundant free-living ASVs occurring in the particle-associated fraction, we also did observe many typical sediment or particle-associated organisms throughout the top 25 ranks and across the majority of sites. These included ASV21 (unk. *Saprospiraceae*), ASV33 (*Pseudoalteromonas* sp.), ASV52 (unk. *Sphingobacteriales*), and ASV56 (unk. *Verrucomicrobiae*) (Fig. 3; Fig. S1) (58, 62–65). Like in the free-living fraction, many particle-associated ASVs had differential relative abundances across salinity. We observed ASVs such as ASV17 (*Sediminibacterium* sp.), ASV77 and ASV114 (unk. *Kapabacteriales*), and ASV116 (unk. *Pedosphaeraceae*) enriched at freshwater sites, with ASV84 (unk. *Cryomorphaceae*) more abundant in high brackish waters (Fig. 4B; Table S1: pairwise), supporting previous findings (66). Notably, the RS62 marine group of Betaproteobacteria (ASV25) was significantly enriched (max relative abundance = 0.30) in the particle-associated fraction at site Freshwater City (FWC) in April 2016 (FWC2) (Fig. 3B). Outside of FWC2, it maintained a low relative abundance (median relative abundance = 0, average relative abundance = 0.01) (Fig. 3B; Table S1: >2.7 μm community). It is difficult to distinguish if the abundance of RS62 was correlated to an unmeasured phytoplankton bloom or if it was a remnant population flushed in during tidal exchange. RS62 taxa occurred in high relative abundances in the planktonic fractions in the Pearl River estuary system (63) but had strong associations with phytoplankton blooms elsewhere (67).

## Brackish communities

Although salinity was the strongest factor separating the microbial communities in our samples (Fig. 2; Table S1: envfit), many taxa occurred across all salinities with peak relative abundances at sites with brackish conditions (Fig. 3 and 5; Fig. S1). This euryhaline distribution can signify a specifically brackish-water-adapted community. We observed many taxa with high relative abundances and frequencies that were also poorly correlated to salinity [Spearman rank correlation coefficient (rho) near 0] (Fig. 6; Fig. S5), supporting the conclusion of a brackish-adapted core microbiome in the nGoM estuaries. Some microorganisms in this brackish group maintained high median relative abundances and occurred at >75% of the sites sampled over 3 years (Fig. 6) and, therefore, likely represent autochthonous brackish and/or euryhaline nGoM taxa. Abundant brackish taxa from both fractions included SAR11 subclade IIIa.3 (ASV5), OM43 (ASV7), Aegean-169 (ASV24), and SAR11 subclade II (ASV9) (Fig. 6; Table S1: Spearman rank correlations), as well as unk. *Saprospiraceae* (ASV645) and unk.

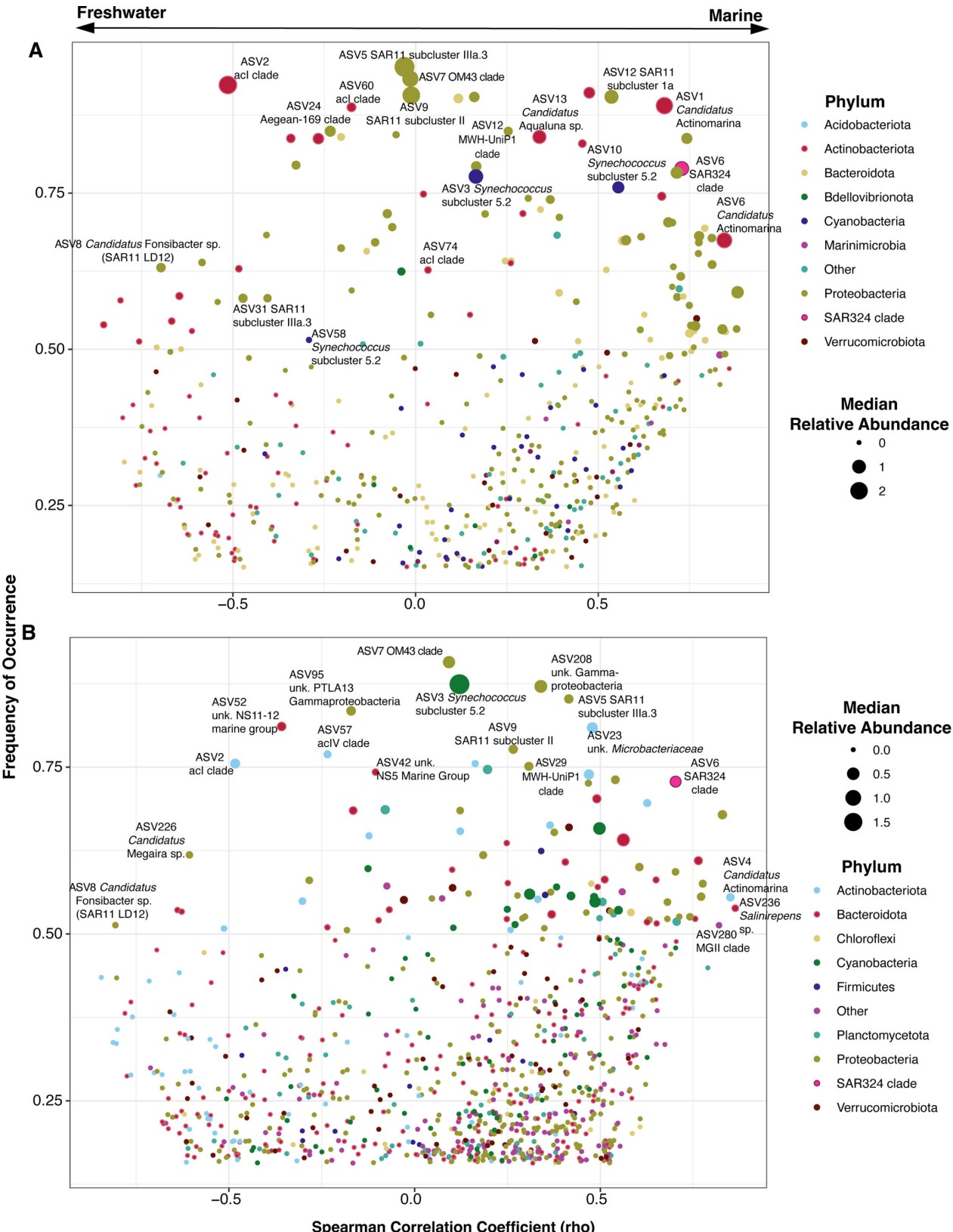

**FIG 6** Two-sided Spearman's rank correlation coefficient (rho) of free-living (A) and particle-associated (B) fraction ASVs according to their frequency of occurrence across all sites. Only ASVs that appear in at least four sites are displayed. ASVs are color-coded by phylum, and the size of the dot corresponds to the median relative abundance for each ASV. The bi-directional arrow above the dot plots provides an orientation of correlations to potential salinity associations. Negative rho values correspond to potential freshwater associations. Positive rho values correspond to potential marine associations.

Gammaproteobacterium PLTA13 (ASV95) in the particle-associated fraction. We also note that SAR11 subclade IIIa.3 had an ASV that was more associated with lower salinity (ASV31; Fig. 5A), and thus, salinity-based ecotypes may occur at the ASV level within some clades. Comparatively, the particle-associated fraction had fewer autochthonous brackish ASVs, but a core group was still present (Fig. 6B; Table S1: Spearman rank correlations).

## DISCUSSION

Estuarine ecosystems have distinct and heterogeneous microbial community structures compared to open ocean marine environments due to the contributions from both riverine and marine sources (25, 45, 68–70). Although this observation has been reproduced in several locations (25, 45, 68–70), estuarine and coastal systems are drastically undersampled for microbial observation compared to open ocean environments, and the extent to which brackish microbiomes from these diverse estuary communities overlap is still unknown. This leaves a substantial knowledge gap that is critical for understanding global biogeochemical cycling because coastal communities can perform a disproportionate amount of turnover, for example, with carbon cycling, compared to their open ocean counterparts (71, 72). Furthermore, the interconnected coastal and estuarine ecosystems are under increasing pressure due to land loss and sea level rise (4, 35). The samples we collected across the nGoM Louisiana coastline represent a diverse collection of environments with notable differences in salinity and other environmental regimes compared to the more stable open ocean. Our research contributes valuable insights into the microbial ecology of estuarine and coastal ecosystems, emphasizing the need for further studies to elucidate population-level diversity, functional roles, and temporal variations in these dynamic environments. These data will serve as both a baseline and an important resource for future researchers investigating the microbial communities found across the coastal nGoM and estuaries globally.

Salinity has repeatedly been observed as the strongest evolutionary and ecological selective factor across aquatic clades and ecosystems (8–10, 15). The fluctuating salinity gradients within estuaries lead to transitory freshwater and marine lineages with fewer well-adapted brackish community members (70), generalized in the Remane curve of species richness across the salinity spectrum (16, 17, 73). Within the nGoM estuaries, a higher number of taxa were strongly correlated to marine environments than intermediate salinity (Fig. S5). However, only 2 of the 47 sites sampled represented freshwater (salinity <0.5; ARD and Swamp), making it difficult to assess the fit of our data to the full Remane curve (Table S1: sampling sites). We did observe that typical freshwater taxa such as acI and *Candidatus* Fonsibacter sp. (SAR11 LD12) taxa were strongly correlated with salinities below 5 (Fig. 3A and 6A), while typical marine taxa were more strongly correlated with sites with salinities above 20 (Fig. 3A and 6A), highlighting the freshwater marine divide and the dynamic nature of coastal estuaries found in other studies (3, 14, 22, 45, 74, 75).

Sites such as Bay of Batiste (BBAT) and Freshwater City (FWC) had large variations in salinity (oligohaline to polyhaline) that were reflected in community composition (Fig. 2), such as switching taxa from Actinobacteria acI to OM1 or SAR11 subclade IIIa to subclade I. Shifts in these distinct communities would have important cellular energetics implications such as change in central metabolism: Embden–Meyerhof–Parnas glycolysis (freshwater) versus Entner–Doudoroff pathway (marine), loss of C1 metabolism (freshwater), and the reliance on *de novo* synthesis (freshwater) rather than uptake (marine) of many important amino acids, osmolytes, and other compounds (6, 8, 14, 27). Changes in these energetics and metabolisms could fundamentally alter carbon cycling and other nutrient availability. Alterations to these processes can lead to differences in carbon transformation or functional capacity and cascading effects in food web dynamics and nutrient cycling across the ecosystem. Therefore, considering the variability in energetics alongside nutrient availability over time and space is critical for

effective future conservation and management strategies under future climate and sea level rise scenarios.

Taxa like SAR11 subclade IIIa, OM43 clade, and *Synechococcus* subcluster 5.2 have been previously established as core members of the brackish-water microbiome (6, 13, 23–25, 27, 45, 76). Our community analysis fortified these assignments and highlighted the need to include other taxa such as SAR11 subclade II, the acI C2 tribe, SAR324, the MWH-UniP1 clade (ASV29), and specific ASVs within poorly classified groups like unk. *Flavobacteriales* spp. (ASV42, ASV94) and unk. *Planctomycetota* (ASV92), among core brackish-water members (Fig. 2, 4, and 6). Many of these groups are typically considered "marine" or "freshwater" taxa. Nevertheless, in addition to the nGoM, these taxa have been observed in brackish environments such as the Baltic Sea, Chilika Lagoon (Bay of Bengal), and the Chesapeake and San Francisco Bays (3, 22, 25, 61, 70, 77, 78). Thus, our study expands the known biogeography and salinity ranges of these organisms and provides additional evidence of taxa within these clades with a potential preference for brackish environments, not just low salinity or marine.

SAR11 subclade II is found predominantly in temperate open oceans during deep mixing events (46) and in oxygen minimum zones (79). However, this subclade was also recently established as highly abundant in brackish environments (27). Phylogenetically, the two abundant brackish subclade II ASVs (ASV9 and ASV119) formed a unique cluster of nGoM taxa and were sister to taxa found in the San Francisco and Chesapeake Bays (Fig. S3). Our results support the hypothesis that a subset of the SAR11 subclade II is brackish-adapted with potentially unique distributions, similar to other ecotypes within SAR11, such as subclade IIIa (27, 80). While salinity transitions are thought to be rare within a group (9, 15), the distribution and population size of the SAR11 clade may have facilitated multiple transitions to brackish intermediaries within the different subclades. Moreover, whether the nGoM cluster of ASVs represents a unique tribe of brackish-adapted ecotypes in subclade II or is part of a large cluster of brackish organisms is still unknown but highlights the missing diversity within an important aquatic group (27, 80) and the need for additional in-depth sampling of estuary ecosystems.

While the majority of ASVs from the acI Actinobacteria were strongly correlated with low salinity and freshwater environments, two ASVs (ASV60 and ASV74) were uncorrelated with salinity. Unlike other acI freshwater ASVs, ASV60 and ASV74 both had maximum relative abundance at intermediate salinities, suggesting these taxa may represent a brackish-adapted subclade within the acI clade. Phylogenetically, ASV74 clusters with other nGoM taxa sister to taxa within the acI-C2 tribe, while ASV60 clusters with other nGoM ASVs as an early diverging member within the acI clade (Fig. S4). Previous studies have observed taxa within the "freshwater" acI at salinities of up to 14 (25, 81, 82). We provide strong evidence of taxa within acI-C with unique distributions at brackish salinities across multiple GoM estuaries, suggesting that members of the acI clade may be core members of both fresh and brackish environments (Table S1: free-living RA). Taken together with the distribution of other microbial clades, our data suggest that brackish environments are important diversification hotspots for aquatic microorganisms that may result from continuous exposure to fluctuating salinity gradients.

The SAR324 clade [previously known as Marine Group B, as a member of the Deltaproteobacteria, and most recently as a candidate phylum (83) is found throughout the global oceans but predominantly in bathypelagic waters (84–86). Nevertheless, the SAR324 clade has also been observed as a significant member of planktonic estuarine and coastal communities, for example, in the Amazon River plume and other estuaries and coastal sediment (58, 85, 86). In the Gulf of Mexico, SAR324 was dispersed across the Louisiana continental shelf, albeit at low relative abundances (30, 87). Within our coastal nGoM samples, the SAR324 clade was the 12th most abundant taxon (ASV6) and was present in all samples with salinity above 2 (Fig. 3; Fig. S1), peaking in relative abundance between salinities of 18 and 23 before sharply decreasing (Table S1: free-living RA; Fig. S1). ASV6 and others (e.g., ASV328, ASV653, and ASV690) classified as SAR324 imply a brackish-adapted subclade within SAR324 that warrants further investigation (Table

S1: free-living RA). Analysis of cross-biome transitions found that genome plasticity may be a hallmark feature of brackish-adapted organisms (6). Indeed, members of the SAR324 clade have diverse metabolic capabilities, allowing them to exploit various energy sources and substrates (6, 86), which may have facilitated their ecological and evolutionary transition to brackish environments. However, given our limited taxonomic and metabolic clarity with amplicon sequencing, further genomic and physiological studies are needed to place these taxa within the associated clades and test their salinity preferences. Future studies should incorporate more estuarine samples to help further delineate the global brackish microbiome and investigate the functional diversity and processes that led to its differentiation.

Without physiological testing, it is difficult to ascertain the specific salinity preference of a taxon, e.g., fresh, brackish, or marine. An organism may be euryhaline but not brackish-adapted, despite both euryhaline and brackish-adapted organisms sharing adaptive salinity responses. For instance, while cultivars from SAR11 subclade IIIa.3 and IIIa.1 are both euryhaline, the IIIa.3 cultivar had optimal growth in mesohaline salinities, while IIIa.1 grew similarly across salinities above 10 (27). Similarly, two coastal isolates within the euryhaline *Synechococcus* subcluster 5.2 CB4 clade had varying growth optima driven by differences in their capacity to regulate osmolyte production and metabolic capacity (20). It is possible that the DNA detected could result from dead cells, which is known to complicate microbial diversity estimates (88, 89). Therefore, it is important for future studies to examine the physiological characteristics of taxa within the pan-brackish microbiome to unveil the genomic underpinnings that delineate transient cells and euryhaline and brackish-adapted taxa.

A major and growing concern for many stakeholders in freshwater and coastal environments is harmful algal blooms (HABs). Analogous to red tide HABs in marine environments, cyanobacterial HABs (cyanoHABs) occur in freshwater and can lead to fish kills, contaminated drinking water, and economic and ecological loss (90–94). Within the particle-associated fraction, two taxa, ASV77 and ASV114 (unk. *Kapabacteriales),* were significantly correlated with low salinity (Fig. 4B; Table S1: pairwise), were the second and fourth most abundant taxa at Lake Martin, and shared high sequence identity to numerous Operational Taxonomic Unit (OTUs) (BLASTn hits >99%) from eutrophic freshwater lakes and HABs (95), particularly *Microcystis* blooms (96). Furthermore, a recently assembled metagenome-assembled genome from the *Kapabacteriales* originated from a culture of the HAB-causing cyanobacterium *Dolichospermum* (97). Thus, at least some *Kapabacteriales* are closely associated with cyanoHABs and may be indicators of such. Within both size fractions, we found numerous other ASVs associated with cyanoHABs (e.g., *Microcystis* and *Planktothrix*) at freshwater coastal (ARD) and inland (Lake Martin) sites, albeit at lower relative abundance than the *Kapabacteriales*-associated ASVs. Although no cyanoHABs were reported during the time of sampling at these sites, the presence of these cyanobacteria and other associated bacteria highlights the potential for cyanoHABs to impact these coastal locations, particularly at lower salinity sites like ARD (93).

## Conclusion

The dynamic interplay between marine, freshwater, and terrestrial inputs in the biologically productive, yet environmentally sensitive, northern Gulf of Mexico estuaries and coastal zones highlights the necessity of understanding the microbial communities underpinning these economically and ecologically vital ecosystems (31, 32, 38, 98). Our study makes an important contribution to the aquatic microbial ecology of these understudied habitats by contributing to the growing knowledge of the globally distributed, core brackish microbiome, composed of members from important aquatic clades such as OM43, *Synechococcus*, and SAR11 (6, 13, 21, 23–25, 27, 70) that should be expanded to include taxa from groups such as acI Actinobacteria, MWH-UniP1 Betaproteobacteria, SAR324, and SAR11 subclade II. Moreover, this study highlights the potential for estuaries to house important biodiversity, such as the observed brackish cluster

within the freshwater acI clade. Future research should incorporate time series data, collection of genomic information, and new culturing efforts to help resolve population-level diversity, function, and temporal variation of the endogenous brackish community in these dynamic environments.

## MATERIALS AND METHODS

### Sample collection

Surface water (<1 m) was collected at nine different sites. Sampling was done as previously described (14, 43, 44). Briefly, duplicate water samples were sequentially filtered through a 2.7 µm GF/D filter (Whatman, UK) and 0.22 µm Sterivex filter (Millipore, USA) until 120 mL was passed, or the filters clogged, using a handheld 60 mL syringe (Becton-Dickinson, USA). We refer to fractions collected on the 2.7 and 0.22 µm filters as particle-associated (>2.7 µm) and free-living (0.22–2.7 µm) fractions, respectively. Sterivex filtrate was analyzed for $SiOH_4$, $NO_3^{-2}$, $NO_2^-$, $NH_4^+$, and $PO_4^{3-}$ at the University of Washington Marine Chemistry Laboratory (http://www.ocean.washington.edu/story/Marine+Chemistry+Laboratory). Filters were immediately placed on ice, transferred to the lab (maximum of 3 h on ice), and frozen at −20°C until further processing. Baseline water conditions of temperature, salinity, pH, and dissolved oxygen were measured using a handheld YSI 556 multiprobe system (YSI Inc., USA). All site locations (latitude and longitude), water chemistry, water conditions, and sampling dates can be found in Table S1 in the "Sampling sites" tab.

### nGoM coastal sites

Sites were sampled once a year for 3 years, except for Terrebonne Bay, which was sampled twice. The sites sampled were Lake Borgne (LKB, Shell Beach, LA) from the Pontchartrain watershed, Bay Pomme d'Or (JLB, Buras, LA) from the Barataria watershed, Terrebonne Bay (TBON, Cocodrie, LA) from the Terrebonne watershed, Atchafalaya River Delta (ARD, Franklin, LA) from the Vermilion-Teche/Atchafalaya watershed, Freshwater City (FWC, Kaplan, LA) from Mermentau watershed, and Calcasieu Jetties (CJ, Cameron, LA) from the Calcasieu Watershed (Table 1). The sample collection was done previously as part of the 3-year cultivation campaign between September 2014 and February 2017 (43).

### Estuarine nGoM sites

Two sites were sampled once roughly every 2 months for 5 months between July 2015 and November 2015. The sites sampled were Sabine Wetlands (Sabine, Cameron, LA) from the Calcasieu watershed and Bay Batiste (BBAT, Port Sulphur, LA) from the Barataria watershed (Table 1). Water was collected from the surface water (<1 m) and filtered immediately on-site as described above.

### Lake Martin

Lake Martin (Swamp, Breaux Bridge, LA) from the Vermilion-Teche watershed was sampled as an inland lake representative in November 2014 (Table 1). Water was collected from the surface(<1 m) and filtered immediately on-site as described above.

### DNA extraction and sequencing

In total, 96 samples were collected, extracted for DNA, and sequenced. All DNA was extracted and sequenced as previously described (44). Briefly, DNA from both size fractions was extracted using the MoBIO PowerWater DNA kit (MoBIO, USA) and quantified using the Qubit high-sensitivity dsDNA assay kit (ThermoFisher, USA). DNA was frozen at −20°C until sequencing. DNA was sequenced targeting the 16S rRNA gene

V4 region with the 515F-806R primer set at Argonne National Laboratory using Illumina MiSeq 2 × 250 bp paired-end reads (99, 100).

## Data analyses

In total, 9,655,001 raw reads were obtained from sequencing. Data were curated and processed in R (v4.2.2) with the package *DADA2* (v1.21.0) following their published protocol (101). Briefly, data were preprocessed for quality and length using *filterAndTrim* with the flags truncLen = c(240,160), maxN = 0, maxEE = c(2,2), truncQ = 2, rm.phix = TRUE, and compress = TRUE. Using the error rates calculated from the *learnErrors,* unique sequences were determined, and the resulting reads were merged using *mergePairs*. Sequences shorter than 250 and larger than 256 were removed. Chimeras were removed using the tool *removeBimeraDenovopoor* with the method flag "consensus." Taxonomy was assigned using the Silva v138 training set and the *assignTaxonomy* command (102).

Because of poor sequencing depth (<1,000 sequences), two samples from the >2.7 µm fraction from the Bay Batiste (BBAT1A PRE and BBAT1B PRE) were removed from all downstream analyses, resulting in 94 remaining samples. Samples ranged in the total number of sequences from 4,785 to 152,624. ASVs assigned to chloroplast (order), mitochondria (family), eurkaryote (kingdom), and unknown (kingdom) were removed manually. Data were processed using the PhyloSeq package in the R (v4.0.2) statistical environment following a protocol similar to those previously published (14, 44, 103). Our modified PhyloSeq script is available on our GitHub repository: https://github.com/thrash-lab/Modified-Phyloseq. After filtering, the data set contained 11,791 unique ASVs and 7,341 ASVs after rarefying with the function *rarefy_even_depth* (Table S1). Alpha diversity was calculated on unrarefied data using the function *plot_richness*. Statistical comparisons between size fraction alpha diversity were made using the function *geomsignif* from the package ggsignif (104). Beta diversity between sites was examined using Bray–Curtis distances via ordination with NMDS. Measured environmental parameters were normalized using the R function *scale* before downstream analyses. *envfit* was used to interpret which environmental parameters were significantly contributing to the NMDS ordination by fitting vectors of significant variables onto the beta diversity-based distance matrix (105). Relative abundances of an ASV from each sample were calculated and averaged between biological duplicates as previously published (44, 103).

## acI and SAR11 taxonomic classification

To better resolve the taxonomic and ecotype designation of acI and SAR11 clade-classified ASVs, ASVs were clustered with near full-length 16S rRNA gene sequences. Sequences were obtained from reference (27) for SAR11 and reference (82) and TaxAss for acI (27, 82, 106). ASV phylogenetic placement was inferred as previously described (27, 44). ASVs with improved taxonomic resolutions were updated with new clade designations (Fig. S3 and S4).

## Statistical analyses

All statistical analyses were performed in R (v4.2.2). Water chemistry for principal components analysis was normalized using root mean square with the command *scale*() and then conducted using the *rda*() function. Bacterial relative abundances that varied between salinity groups were identified with the Kruskal–Wallis test with Benjamini–Hochberg correction for testing all identified ASVs, followed by pairwise Wilcoxon comparisons with Benjamini–Hochberg correction (107). We employed two-sided Spearman's rank correlation to determine the relationship between taxa relative abundance and salinity. To control for the false discovery rate, *P*-values were adjusted using the Benjamini and Hochberg method (108) using *p.adjust*() as a flag. Our modified scripts are available on the GitHub repository: https://github.com/theaquatic-microbiologylab/Henson_Thrash_CoastalnGoM.

## ACKNOWLEDGMENTS

A portion of this research was conducted with high-performance computing resources provided by Louisiana State University (http://www.hpc.lsu.edu), and the authors acknowledge the Center for Advanced Research Computing (CARC) at the University of Southern California for providing computing resources that have contributed to the research results reported within this publication (URL: https://carc.usc.edu).

This work was funded by the Louisiana Board of Regents (Board of Regents) (LEQSF[2014-2017]-RDA-06), the Louisiana State University Department of Biological Sciences, a National Academies of Science, Engineering, and Medicine Gulf Research Program Early Career Research Fellowship, and a Simons Investigator in Aquatic Microbial Ecology Award from the Simons Foundation to J.C.T.; the Dornsife College of Letters, Arts, and Sciences at the University of Southern California; the Department of Biological Sciences and College of Liberal Arts and Sciences at Northern Illinois University; the Department of Geophysical Sciences at the University of Chicago; and a LEEC grant from the Louisiana Department of Fish and Wildlife, Lerner Gray, grant from the American Natural History Museum, and a Simons Foundation Marine Microbiology Postdoctoral Fellowship to M.W.H.

## AUTHOR AFFILIATIONS

[1]Department of Biological Sciences, Northern University, DeKalb, Illinois, USA
[2]Department of Biological Sciences, University of Southern California, Los Angeles, California, USA

## AUTHOR ORCIDs

Michael W. Henson  http://orcid.org/0000-0002-4351-797X
J. Cameron Thrash  http://orcid.org/0000-0003-0896-9986

## FUNDING

| Funder | Grant(s) | Author(s) |
| --- | --- | --- |
| Simons Foundation (SF) | Simons Investigator in Aquatic Microbial Ecology Award | J. Cameron Thrash |
| Simons Foundation (SF) | Marine Microbiology Postdoctoral Fellowship | Michael W. Henson |
| Louisiana Board of Regents (LBR) | LEQSF[2014-2017]-RDA-06 | J. Cameron Thrash |
| American Museum of Natural History (AMNH) | Lerner Gray grant | Michael W. Henson |
| National Academies of Sciences, Engineering, and Medicine (NASEM) | Gulf Research Program Early Career Research Fellowship | J. Cameron Thrash |
| Louisiana Department of Fish and Wildlife | LEEC Grant | Michael W. Henson |

## AUTHOR CONTRIBUTIONS

Michael W. Henson, Data curation, Formal analysis, Funding acquisition, Investigation, Methodology, Project administration, Visualization, Writing – original draft, Writing – review and editing | J. Cameron Thrash, Conceptualization, Data curation, Formal analysis, Funding acquisition, Methodology, Project administration, Supervision, Writing – original draft, Writing – review and editing

## DATA AVAILABILITY

The free-living fraction (0.22–2.7 µm) raw read sequences are available at the Sequence Read Archive (SRA) with accession numbers SRR6235382–SRR6235415 as previously published (14, 44). All other free-living fraction (0.22–2.7 µm) raw read sequences are available at the SRA with accession numbers SRS1840441 ‑–SRS1840447. All particle-associated fraction (> 2.7 µm) raw read sequences are available at the SRA with accession numbers SRR18184264–SRR18184311.

## ADDITIONAL FILES

The following material is available online.

### Supplemental Material

**Figure S1 (mSystems01318-23-s0001.pdf).** Top 25 relative abundance curve.
**Figure S2 (mSystems01318-23-s0002.pdf).** Alpha diversity of samples.
**Figure S3 (mSystems01318-23-s0003.pdf).** Phylogenetic tree of SAR11 with nGoM amplicons.
**Figure S4 (mSystems01318-23-s0004.pdf).** Phylogenetic tree of acI clade with nGoM amplicons.
**Figure S5 (mSystems01318-23-s0005.pdf).** Frequency plot of rho values (Spearman rank correlation) from salinity analysis.
**Table S1 (mSystems01318-23-s0006.xlsx).** Supplemental data and results.

### Open Peer Review

**PEER REVIEW HISTORY (review-history.pdf).** An accounting of the reviewer comments and feedback.

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
