## [Reviewer comments · mSystems]

Microbial ecology of northern Gulf of Mexico estuarine waters

Michael Henson and J. Cameron Thrash

Corresponding Author(s): Michael Henson, Northern Illinois University

Review Timeline:

Submission Date:	December 6, 2023
Editorial Decision:	February 15, 2024
Revision Received:	May 24, 2024
Accepted:	June 19, 2024

Editor: Ryan Newton

Reviewer(s): Disclosure of reviewer identity is with reference to reviewer comments included in decision letter(s). The following individuals involved in review of your submission have agreed to reveal their identity: Bradley B. Tolar (Reviewer #2); Anirban Chakraborty (Reviewer #3)

Transaction Report:

DOI: <https://doi.org/10.1128/msystems.01318-23>

Re: mSystems01318-23 (**Microbial ecology of coastal northern Gulf of Mexico waters**)

Dear Dr. Michael Henson:

Thank you for sending your manuscript to mSystems. Three reviewers have now provided comments. The reviewers generally agree that the manuscript presents a nicely compiled and well described study on the microbial communities of an estuarine system. The reviewers all provided well thought out reviews and note in their comments ways the manuscript can be improved in a revision. Please look at these comments closely and provide your response and a revised manuscript. Among the comments, the reviewers note a few technical aspects that should be addressed in the revision as related to sampling, sample processing, and analysis of compositional data. Clarifying the described thought processes and describing potential impacts on interpretation from the technical choices are important points to consider in your revision.

Revision Guidelines

Sincerely,
Ryan Newton
Editor
mSystems

Reviewer #1 (Comments for the Author):

In their manuscript, Microbial ecology of coastal northern Gulf of Mexico waters, Henson and Thrash characterize bacterial communities in estuaries along a gradient of salinities using 16S rRNA metabarcoding sequencing. The authors argue that this study is particularly significant due to a relative lack of similar studies in estuarine ecosystems compared to other ecosystems. Moreover, estuaries are ecologically important while being particularly sensitive to anthropogenic impacts. This study is framed as a necessary and important baseline that will allow the detection of changes in microbial communities in the future. The main findings are that the microbial community compositions significantly differ by salinity and that clades typically associated with marine or freshwater were enriched in brackish water. The manuscript is well-written and has quality data visualizations. The main issues to be addressed concern whether statements made about "habitat preference" can be supported with compositional data and replication/study design. Please see the detailed comments below.

Major Issues:

1- The fact that amplicon data are compositional data needs to be addressed more directly. How does that affect the interpretation of your results? Is it possible to make conclusions about habitat preference?

2- Six sites were sampled once a year for three years, two sites were sampled three times over 5 months, and one site was sampled only once. How were the time of year, tide, and hydrological setting (recent storms/rain) considered in the analysis and interpretation of the results?

3- There did not seem to be any comparison to nearby marine communities. Is there data available that could be used for this? There was sampling of a freshwater lake to provide a comparison, it seems like an oversight to exclude a marine comparison.

4- The discussion would benefit from subheadings

5- Figure legends need more details and context - they cannot stand alone as currently written.

Minor Issues:

First paragraph of the introduction could be rewritten/expanded and improved. For example, the last sentence mentions the Remane curve, but this is not really explained until the discussion. The first paragraph does not set up the problem being addressed in the manuscript.

Line 96: "elevated land loss" is a difficult phrase to understand and may cause readers to back track and reread the sentence. I suggest rewriting this hook sentence for readability and impact.

Line 104: It is unclear what the "respectively" clause is referring to here.

Line 115: "The northern Gulf of Mexico (nGoM) provides an excellent coastal/estuarine study system because of its economic and ecological value." - I would argue that economic and ecological value do not necessarily make an excellent study system, but instead make a relevant or important study system. These attributes are the answers to "why here", but I would recommend adding in some specific values such as fisheries supported, incomes, coastal population, etc. The following sentence describing the system (two rivers, interconnected wetlands, fluctuating salinity and nutrient conditions) speaks more to why it is an excellent study system: gradients for comparison, etc.

Line 119: Maybe nitpicky, but I don't think the nGoM loses land since it is a body of water. Perhaps nGoM estuaries or wetlands lose land?

Line 125: "separate from these stressors" - reword

Line 131: "We quantified the particle-associated and planktonic microbial communities as well as the associated water chemistry." - Amplicon data is not truly quantitative. I think this sentence is misleading.

Line 156: values → variables

Line 161: non-metric multidimensional scaling

Line 163: high brackish should be > 12 rather than < 12 (correct elsewhere, typo here)

Line 165: ANOSIM R= should be R2=

Line 183: I think this is the first time in the manuscript that a change in relative abundance is directly equated to habitat preference. This is a really hard assertion to make or justify. If an ASV has a higher abundance in one sample than another, it does not necessarily mean that it is more abundant in the first sample. Instead, another ASV may be more abundant in the

second sample, making it seem like the first ASV is less abundant by causing it to make up a smaller proportion of the total sample. I am not implying that the authors have not considered the compositionality of amplicon data. However, I do think it needs to be given more consideration/discussion before making statements about habitat preference.

Line 186: abundance → relative abundance. I think it's important to always stick with relative abundance, even if it can feel a bit redundant.

Line 220: were different volumes filtered during different sampling events?

221: there is a double space between researchers and should

255: Concerning brackish communities, it would be helpful if the authors included more information about the sites regarding tidal flushing, freshwater input, and daily salinity cycles. If there is a lot of marine or freshwater entering the estuary, bacteria may be constantly introduced from these two habitats. They may persist over short terms or the authors may be detecting DNA from dead cells or environmental DNA.

280: obervation → observations ?

286: worsening pressure → increasing pressure?

297-300: run-on (but important) sentence. introduction/description of the Remane curve could be moved to the introduction. This could also be much more explicit ... Following sentences say that there are typical freshwater taxa in low salinity locations and typical marine taxa in high salinity locations - how do brackish ecosystems fit in? Also, what does the Remane curve actually reflect? Richness? Are there more "species" expected at the extremes and less in the intermediate? Is that what was observed?

Line 305: FWC is defined here, but used earlier without definition.

Line 312-314: this sentence reads as fragments (comma splice or missed oxford comma?) - suggested to revise this sentence

Line 316 - I didn't quite catch the connection between energetics and management - the implications of changing energetics should be laid out more explicitly

Line 322: typo, ASVis → ASVs

Line 345: maximum abundance → maximum relative abundance

Line 368-371: awkward phrasing, suggested revision

Line 358-375: the paragraph about SAR324 feels out of order, the previous paragraph describes other potentially brackish-adapted clades and concludes with a disclaimer describing the shortcomings of amplicon sequencing. The SAR324 information could be incorporated into the previous paragraph, or the disclaimer could be moved down.

377 - Euryhaline is defined here, but is already used above. The definition could be in the introduction or at first use.

Line 393: "blasted to" is used here to indicate high nucleotide % identity in BLASTn searches. Generally, blast as a verb refers to doing a BLAST search, not necessarily matching. I recommend rephrasing these results with clearer language.

Line 396: *Kapabacteriales* was italicized in the previous line.

Line 408-411: run on

411: data contributes → data contribute

Line 417: states that future research should incorporate time series data. There is a temporal element to the data collected here (3 years), but this is not discussed in the results or discussion sections?

471: Silva V132 → SILVA v132

477: PhyloSeq program → PhyloSeq package

510: what are iTag sequences? Are these the raw reads or the ASVs? May be better to avoid new jargon?

Table 1: It is unclear in the table what published/unpublished is or why it is included. The sampling dates show that the samples were not collected in the same month each year-do you expect seasonality? Were they collected at the same time of day /

same tide? How would incoming or outgoing tides impact your results?

Figure 1A: Map needs more labeling and a scale. An inset showing where this map sits in a larger map would be helpful. It would also be extremely helpful if the map included rivers with the river names labeled. Also, where there are three points for each site, the shape could reflect the year or years could be labeled. Include full names of sampling locations in the legend.

Figures 2 & 4, high salinity is mislabeled as orange in the legends

Figures 3 & 5 colors are very difficult to distinguish (especially the purples)

Figure 4: Differential abundance testing - is this pairwise? What package did you use? DESeq2?
Polygon → hexagon (I believe all multisided shapes are polygons, including stars and rectangles)

Figure 5: Revise "across with salinity". What nonlinear regression method did you use?

Reviewer #2 (Comments for the Author):

This work represents a survey of the microbial communities of the Northern Gulf of Mexico (nGOM) along the Louisiana (LA) coastline, a region that has been fairly overlooked especially in microbial ecology (though, coastal estuaries in general are also understudied). This study sampled eight coastal sites and one inland freshwater site and determined the microbial communities by sequencing two sized fractions using Illumina 16S amplicon sequencing with 515F-806R primers. The major finding of this work is that microbes of the nGOM on the LA coastline follow a distribution driven mainly by salinity as predicated by the Reman curve, and that particle size and thus turbidity appear to play an important role. The other finding is that there is a common, core brackish microbiome between many global estuaries. This is significant as most recent studies of the nGOM focused on oil-spill recovery and other stressors, and thus this dataset adds to our understanding of the "normal" microbiome of nGOM. Overall this work is well-written and easy to follow, and moves the field forward in our understanding of these systems. However, there are a few areas of the paper that could be strengthened or clarified as outlined in the attached file.

Reviewer #3 (Comments for the Author):

In this manuscript, the authors present a robust dataset of high throughput 16S rRNA gene amplicon sequencing survey used to investigate spatiotemporal patterns of the microbiome diversity in the northern Gulf of Mexico (nGoM) waters along the Louisiana coast. Nine sampling sites represented a wide range of salinity from freshwater inland to high-brackish coastal waters. Furthermore, particle-associated and planktonic communities were separated at the time of sampling by serial filtration and were investigated separately. Habitat preference driven primarily by salinity of many known freshwater and marine clades were observed.

Overall, this study provides important microbial community baseline data focused on coastal habitats. Below are some suggestions for improvement.

Specific comments:

What was the overall distribution of bacteria and archaea across samples? A brief description would be helpful, perhaps aided by a supplementary figure showing overall community composition at phylum level.

Lines 132 - 137: "Six sites ... sampled only once (Tables 1 and S1)." - move these lines to the first paragraph of the Results section.

Fig. 1A: In most figures, three sample groups (based on salinity) were displayed using a defined color pattern. Please consider retaining the same color pattern in Fig. 1A for consistency. Salinity values as a continuous variable can still be shown with varying symbol size.

In Figs 3 and 5 relative abundance was shown as fraction while in Fig. 4 it was shown as percentage. A consistent unit of relative abundance across all figures would be helpful for readers.

Line 425: Incomplete parentheses... "(Becton-Dickinson, USA" Replace "gram-negative phylum" with "phylum of gram-negative bacteria".

Lines 459 - 461: During PCR amplification, how many replicate reactions per sample were conducted?

Line 464: What was the read distribution across samples?

Line 471: Why was the most updated version of SILVA (v138.1) not used for taxonomy assignment?

Lines 510 - 514: Please replace "iTag sequences" with "raw sequence reads".

Lines 591 & 602: Please use a distinct color name (other than orange) for the high-brackish group.

In their manuscript, Microbial ecology of coastal northern Gulf of Mexico waters, Henson and Thrash characterize bacterial communities in estuaries along a gradient of salinities using 16S rRNA metabarcoding sequencing. The authors argue that this study is particularly significant due to a relative lack of similar studies in estuarine ecosystems compared to other ecosystems. Moreover, estuaries are ecologically important while being particularly sensitive to anthropogenic impacts. This study is framed as a necessary and important baseline that will allow the detection of changes in microbial communities in the future. The main findings are that the microbial community compositions significantly differ by salinity and that clades typically associated with marine or freshwater were enriched in brackish water. The manuscript is well-written and has quality data visualizations. The main issues to be addressed concern whether statements made about “habitat preference” can be supported with compositional data and replication/study design. Please see the detailed comments below.

Major Issues:

- 1- The fact that amplicon data are compositional data needs to be addressed more directly. How does that affect the interpretation of your results? Is it possible to make conclusions about habitat preference?
- 2- Six sites were sampled once a year for three years, two sites were sampled three times over 5 months, and one site was sampled only once. How were the time of year, tide, and hydrological setting (recent storms/rain) considered in the analysis and interpretation of the results?
- 3- There did not seem to be any comparison to nearby marine communities. Is there data available that could be used for this? There was sampling of a freshwater lake to provide a comparison, it seems like an oversight to exclude a marine comparison.
- 4- The discussion would benefit from subheadings
- 5- Figure legends need more details and context - they cannot stand alone as currently written.

Minor Issues:

First paragraph of the introduction could be rewritten/expanded and improved. For example, the last sentence mentions the Remane curve, but this is not really explained until the discussion. The first paragraph does not set up the problem being addressed in the manuscript.

Line 96: “elevated land loss” is a difficult phrase to understand and may cause readers to back track and reread the sentence. I suggest rewriting this hook sentence for readability and impact.

Line 104: It is unclear what the “respectively” clause is referring to here.

Line 115: "The northern Gulf of Mexico (nGoM) provides an excellent coastal/estuarine study system because of its economic and ecological value." - I would argue that economic and ecological value do not necessarily make an excellent study system, but instead make a relevant or important study system. These attributes are the answers to "why here", but I would recommend adding in some specific values such as fisheries supported, incomes, coastal population, etc. The following sentence describing the system (two rivers, interconnected wetlands, fluctuating salinity and nutrient conditions) speaks more to why it is an excellent study system: gradients for comparison, etc.

Line 119: Maybe nitpicky, but I don't think the nGoM loses land since it is a body of water. Perhaps nGoM estuaries or wetlands lose land?

Line 125: "separate from these stressors" - reword

Line 131: "We quantified the particle-associated and planktonic microbial communities as well as the associated water chemistry." - Amplicon data is not truly quantitative. I think this sentence is misleading.

Line 156: values → variables

Line 161: non-metric multidimensional scaling

Line 163: high brackish should be > 12 rather than < 12 (correct elsewhere, typo here)

Line 165: ANOSIM R= should be $R^2=$

Line 183: I think this is the first time in the manuscript that a change in relative abundance is directly equated to habitat preference. This is a really hard assertion to make or justify. If an ASV has a higher abundance in one sample than another, it does not necessarily mean that it is more abundant in the first sample. Instead, another ASV may be more abundant in the second sample, making it seem like the first ASV is less abundant by causing it to make up a smaller proportion of the total sample. I am not implying that the authors have not considered the compositionality of amplicon data. However, I do think it needs to be given more consideration/discussion before making statements about habitat preference.

Line 186: abundance → relative abundance. I think it's important to always stick with relative abundance, even if it can feel a bit redundant.

Line 220: were different volumes filtered during different sampling events?

221: there is a double space between researchers and should

255: Concerning brackish communities, it would be helpful if the authors included more information about the sites regarding tidal flushing, freshwater input, and daily salinity cycles. If there is a lot of marine or freshwater entering the estuary, bacteria may be constantly introduced from these two habitats. They may persist over short terms or the authors may be detecting DNA from dead cells or environmental DNA.

280: obervation → observations ?

286: worsening pressure → increasing pressure?

297–300: run-on (but important) sentence. introduction/description of the Remane curve could be moved to the introduction. This could also be much more explicit ... Following sentences say that there are typical freshwater taxa in low salinity locations and typical marine taxa in high salinity locations - how do brackish ecosystems fit in? Also, what does the Remane curve actually reflect? Richness? Are there more “species” expected at the extremes and less in the intermediate? Is that what was observed?

Line 305: FWC is defined here, but used earlier without definition.

Line 312–314: this sentence reads as fragments (comma splice or missed oxford comma?) - suggested to revise this sentence

Line 316 - I didn't quite catch the connection between energetics and management - the implications of changing energetics should be laid out more explicitly

Line 322: typo, ASVis → ASVs

Line 345: maximum abundance → maximum relative abundance

Line 368–371: awkward phrasing, suggested revision

Line 358–375: the paragraph about SAR324 feels out of order, the previous paragraph describes other potentially brackish-adapted clades and concludes with a disclaimer describing the shortcomings of amplicon sequencing. The SAR324 information could be incorporated into the previous paragraph, or the disclaimer could be moved down.

377 - Euryhaline is defined here, but is already used above. The definition could be in the introduction or at first use.

Line 393: “blasted to” is used here to indicate high nucleotide % identity in BLASTn searches. Generally, blast as a verb refers to doing a BLAST search, not necessarily matching. I recommend rephrasing these results with clearer language.

Line 396: Kapabacteriales was italicized in the previous line.

Line 408-411: run on

411: data contributes → data contribute

Line 417: states that future research should incorporate time series data. There is a temporal element to the data collected here (3 years), but this is not discussed in the results or discussion sections?

471: Silva V132 → SILVA v132

477: PhyloSeq program → PhyloSeq package

510: what are iTag sequences? Are these the raw reads or the ASVs? May be better to avoid new jargon?

Table 1: It is unclear in the table what published/unpublished is or why it is included. The sampling dates show that the samples were not collected in the same month each year—do you expect seasonality? Were they collected at the same time of day / same tide? How would incoming or outgoing tides impact your results?

Figure 1A: Map needs more labeling and a scale. An inset showing where this map sits in a larger map would be helpful. It would also be extremely helpful if the map included rivers with the river names labeled. Also, where there are three points for each site, the shape could reflect the year or years could be labeled. Include full names of sampling locations in the legend.

Figures 2 & 4, high salinity is mislabeled as orange in the legends

Figures 3 & 5 colors are very difficult to distinguish (especially the purples)

Figure 4: Differential abundance testing - is this pairwise? What package did you use? DESeq2?

Polygon → hexagon (I believe all multisided shapes are polygons, including stars and rectangles)

Figure 5: Revise “across with salinity”. What nonlinear regression method did you use?

Response to Reviewer Comments for mSystems01318-23

Original comments in black italics. Our responses are in red normal text.

We want to thank all the reviewers for their time and thorough and helpful suggestions throughout the manuscript. Their comments improved the manuscript.

Reviewer #1 (Comments for the Author):

In their manuscript, Microbial ecology of coastal northern Gulf of Mexico waters, Henson and Thrash characterize bacterial communities in estuaries along a gradient of salinities using 16S rRNA metabarcoding. The authors argue that this study is particularly significant due to a relative lack of similar studies in estuarine ecosystems compared to other ecosystems. Moreover, estuaries are ecologically important while being particularly sensitive to anthropogenic impacts. This study is framed as a necessary and important baseline that will allow the detection of changes in microbial communities in the future. The main findings are that the microbial community compositions significantly differ by salinity and that clades typically associated with marine or freshwater were enriched in brackish water. The manuscript is well-written and has quality data visualizations. The main issues to be addressed concern whether statements made about “habitat preference” can be supported with compositional data and replication/study design. Please see the detailed comments below.

Major Issues:

1- The fact that amplicon data are compositional data needs to be addressed more directly. How does that affect the interpretation of your results? Is it possible to make conclusions about habitat preference?

We recognize that the data is compositional in nature and we accounted for this in our choice of analysis methods:

Salinity Correlations: We used Spearman’s correlations between ASVs and salinity. Spearman's correlation is based on the ranks of the data points rather than their actual values. This makes it invariant to the compositional nature of the data, where the absolute abundances are not known, and only the relative abundances are available. Further, microbial abundance data usually follows non-normal distributions, with many zero or low abundance values and a few high abundance taxa. Spearman's correlation analysis is robust to such skewed distributions, as it relies on the ranks rather than the actual values. Lastly, Spearman's correlations can capture both linear and non-linear associations. This is advantageous when analyzing salinity correlations with microbial relative abundance, as the relationships between abundance and salinity might not be linear.

Differential relative abundances between the three salinity groups: We first used the Kruskal-Wallis test to determine taxa with significant pairwise differences, followed by the Wilcoxon test to determine which of the three salinity groups was significantly different. The Kruskal-Wallis and Wilcoxon tests are non-parametric, which makes them suitable for relative abundance data that may not follow a normal distribution or have homogeneous variances. We employed the Benjamini-Hochberg correction to control false discovery rate, which is important when conducting multiple pairwise comparisons to avoid an inflated Type I error rate.

We have also been attentive to relabeling all instances of “abundance” as “relative abundance” to ensure we’re not accidentally suggesting we were measuring absolute abundances.

To address the “habitat preference” issue, we have removed comments about “preference” and exchanged them with “correlations” or “associations” to be sensitive to the nature of relative abundance data, except in the Discussion where we are making recommendations for how to assess preference or discussing potential preference.

2- Six sites were sampled once a year for three years, two sites were sampled three times over 5 months, and one site was sampled only once. How were the time of year, tide, and hydrological setting (recent storms/rain) considered in the analysis and interpretation of the results?

When we first began this project, we hoped we could use sampling time as one of our variables and maybe suss out seasonality. However, the sampling frequencies across sites were too different, and/or we didn’t have adequate numbers for statistical tests, to use temporal information. Tide and hydrological setting were not considered in the interpretation. Without detailed hydrological information about lag times between rain pulses and delivery to our sampling sites, as well as temporal controls before and after such events, we cannot make any assertions about the effect of hydrology on microbial communities. The study was not designed for such analyses. Although not addressed directly, tidal effects at these locations should be minimal because of the mostly diurnal nature of the tides and the low tidal exchange in the region. For example, the tidal range during all of March of this year near our Atchafalaya sampling site spanned only 2 feet (-0.5 – 1.5). Regardless, we did not plan the sampling around tides, and therefore cannot comment on the effect of tides in the study.

3- There did not seem to be any comparison to nearby marine communities. Is there data available that could be used for this? There was sampling of a freshwater lake to provide a comparison, it seems like an oversight to exclude a marine comparison.

We agree with the reviewer that adding marine samples would make a nice comparison. However, we did not take any truly marine samples (beyond the shelf, salinity > 32) at the time of sampling, so we are stuck with the samples we had in hand for our study.

That said, we have included comparisons to the limited number of previous marine nGoM shelf studies in the paper:

Lines 130-134: “Samples from pre-Deepwater Horizon oil spill microbial communities along the continental Louisiana shelf were dominated by Alphaproteobacteria and Bacteroidota, specifically the SAR11 clade (29). In contrast, deeper samples found an increasing abundance of Archaea, specifically *Crenarchaeota* (a.k.a. Thaumarchaeota, Nitrososphaerota) (41).”

Lines 198-202: “At higher salinities (> 12), typical marine clades such as SAR11 subclade I (Alphaproteobacteria) and *Candidatus Actinomarina* (OM1 clade, Actinobacteria) were more abundant than in samples of lower salinity (**Fig. 4A**), supporting previous observations of their high relative abundance in samples from the Louisiana continental shelf with high salinity (29).”

Lines 390-391: “In the Gulf of Mexico, SAR324 was dispersed across the Louisiana continental shelf, albeit at low relative abundances (29, 86).”

4- The discussion would benefit from subheadings

Although we feel subheadings help in the Results because there were discreet analyses and associated findings, we feel breaking up the Discussion with subheadings would impose an artificial separation of ideas that is not helpful to the narrative structure. Therefore, we respectfully would like to maintain the Discussion without subheadings.

5- Figure legends need more details and context - they cannot stand alone as currently written.

We have updated figure legends throughout to include more information. See the comments below for more details.

Minor Issues:

First paragraph of the introduction could be rewritten/expanded and improved. For example, the last sentence mentions the Remane curve, but this is not really explained until the discussion. The first paragraph does not set up the problem being addressed in the manuscript.

We have restructured the first paragraph to improve clarity and readability:

Lines 87-102: “Estuarine environments are highly diverse, interconnected ecosystems that are exposed to strong natural fluctuations in salinity and nutrient availability (1–3). Under future climate scenarios, estuaries are expected to have increased erosion and intrusion of saltwater from sea level rise, degraded water quality from runoff and pollution from severe storms, and increased temperatures (4–7), all of which could fundamentally alter microbial community

composition and thus the processing of nutrients and food web dynamics (8). Fluctuations in salinity from sea-level rise are especially likely to change microbial community structure and metabolic capacity (8–11). This reflects the fact that organisms require unique metabolic and cellular features in marine and freshwater environments (6, 8, 12–15). These important differences are hypothesized to lead to low species diversity at intermediate salinities and high species diversity in freshwater and marine environments, as first visualized in the Remane curve (16). Although the Remane curve was not originally formulated to describe microorganisms, the pattern was found to apply to phytoplankton and other microbes (17). The relationships described in the curve anticipated the now-supported observation of infrequent transitions between marine and freshwater species (5, 11, 14, 85) and the importance of salinity in structuring microbial communities (4–7, 26).”

Line 96: “elevated land loss” is a difficult phrase to understand and may cause readers to back track and reread the sentence. I suggest rewriting this hook sentence for readability and impact.

We have moved this sentence to the introduction. It now reads:

Line 88-92: “Under future climate scenarios, estuaries are expected to have increased erosion and intrusion of saltwater from sea level rise, degraded water quality from runoff and pollution from severe storms, and increased temperatures (4–7), all of which could fundamentally alter microbial community composition and thus the processing of nutrients and food web dynamics (8).”

Line 104: It is unclear what the “respectively” clause is referring to here.

Thank you. We have removed “respectively” from this sentence.

Line 115: “The northern Gulf of Mexico (nGoM) provides an excellent coastal/estuarine study system because of its economic and ecological value.” - I would argue that economic and ecological value do not necessarily make an excellent study system, but instead make a relevant or important study system. These attributes are the answers to “why here”, but I would recommend adding in some specific values such as fisheries supported, incomes, coastal population, etc. The following sentence describing the system (two rivers, interconnected wetlands, fluctuating salinity and nutrient conditions) speaks more to why it is an excellent study system: gradients for comparison, etc.

Good points. We have rephrased this portion of the text to more clearly identify the features that make the nGoM an excellent study system:

Lines 119-122: “Influenced by two major rivers -- the Mississippi and Atchafalaya Rivers-- and

a vast network of interconnected wetlands, the northern Gulf of Mexico (nGoM) coastline is subject to continuous fluctuations in environmental conditions such as salinity, nutrients, and turbidity (27–29).”

Line 119: Maybe nitpicky, but I don't think the nGoM loses land since it is a body of water. Perhaps nGoM estuaries or wetlands lose land?

Another good point. We have reworded this to say:

Lines 124-125: “Over the past century, nGoM wetlands have lost an estimated 5000 km² of land due to erosion (34).”

Line 125: “separate from these stressors” - reword

To help clarify this sentence, we have reworded this to say:

Lines 127-130: “However, previous microbiological research in this region has mostly focused on the communities associated with oil spills and eutrophication, and there is much less data on the general microbial ecology from the estuary ecosystems across the nGoM (35–40).”

Line 131: “We quantified the particle-associated and planktonic microbial communities as well as the associated water chemistry.” - Amplicon data is not truly quantitative. I think this sentence is misleading.

We have replaced “quantified” with “assessed” to remove any associations with absolute quantification:

Lines 146-148: “We assessed the particle-associated (> 2.7 μm) and free-living (0.2 - 2.7 μm) fractions of the microbial communities as well as the associated water chemistry.”

Line 156: values → variables

Thank you. We have replaced values with variables.

Line 161: non-metric multidimensional scaling

Thank you. We have replaced “analysis” with “scaling.”

Line 163: high brackish should be > 12 rather than < 12 (correct elsewhere, typo here)

Thank you. We have corrected this error in the text.

Line 165: ANOSIM R= should be R²=

ANOSIM outputs the correlation between variables (R) not the proportion of the variance in the response variable that can be explained by the predictor variable (R²). We have kept R in the text.

Line 183: I think this is the first time in the manuscript that a change in relative abundance is directly equated to habitat preference. This is a really hard assertion to make or justify. If an ASV has a higher abundance in one sample than another, it does not necessarily mean that it is more abundant in the first sample. Instead, another ASV may be more abundant in the second sample, making it seem like the first ASV is less abundant by causing it to make up a smaller proportion of the total sample. I am not implying that the authors have not considered the compositionality of amplicon data. However, I do think it needs to be given more consideration/discussion before making statements about habitat preference.

As part of our larger remodel, we have changed instances of “preference” in our results section to “correlations” or “associations”. We have rephrased this specific sentence to say “correlated to” instead of “preference for”:

Lines 206-210: “Other ASVs with differentiated relative abundances were from the SAR86 (Gammaproteobacteria) and SAR324 clades, which both had higher relative abundance in higher salinity. ASV54 unknown *Holophagaceae* (Acidobacteria) and ASV13 *Candidatus Aqualuna* (Actinobacteria) were correlated with freshwater habitats (**Fig. 4A, Table S1**: free-living RA).”

Moreover, in the Discussion, we mention on Lines 405-416 that there are caveats to this type of study. We hope to continue to study these systems to provide more concrete evidence for our hypotheses.

Line 186: abundance → relative abundance. I think it's important to always stick with relative abundance, even if it can feel a bit redundant.

We have changed all instances of abundance to relative abundance.

Line 220: were different volumes filtered during different sampling events?

120 mL of water was sequentially filtered through the two filters or until clogged. We have included this information in the Methods section:

Lines 454-456: “Briefly, duplicate water samples were sequentially filtered through a 2.7 µm GF/D filter (Whatman, UK) and 0.22 µm Sterivex filter (Millipore, USA) until 120 mL was passed, or the filters clogged, using a handheld 60 mL syringe (Becton-Dickinson, USA).”

221: *there is a double space between researchers and should*

Thank you. The double space was removed.

255: *Concerning brackish communities, it would be helpful if the authors included more information about the sites regarding tidal flushing, freshwater input, and daily salinity cycles. If there is a lot of marine or freshwater entering the estuary, bacteria may be constantly introduced from these two habitats. They may persist over short terms or the authors may be detecting DNA from dead cells or environmental DNA.*

Please see our response to Major Issue 2, above, regarding hydrology and tidal flushing. Regarding the persistence of DNA over time from dead, non-viable, or non-active cells, this is a limitation for all DNA-based studies. Throughout the manuscript, we discuss that further studies on the physiology of these organisms are necessary to confirm our hypotheses (e.g., Lines 400-402, Lines 405-416). Moreover, we are careful to provide additional evidence from other studies and phylogenetic analyses to support our observations. We have also amended our manuscript to include the residual DNA caveat:

Lines 418-420: “Moreover, it is possible that the DNA detected could be a relic from dead or decaying cells, which is known to complicate microbial diversity estimates (98, 99).”

280: *obervation* → *observations* ?

Thank you. Obervation was corrected to “observation.”

286: *worsening pressure* → *increasing pressure*?

Worsening was replaced with “increasing”.

297–300: *run-on (but important) sentence. introduction/description of the Remane curve could be moved to the introduction. This could also be much more explicit ... Following sentences say that there are typical freshwater taxa in low salinity locations and typical marine taxa in high salinity locations - how do brackish ecosystems fit in? Also, what does the Remane curve actually reflect? Richness? Are there more “species” expected at the extremes and less in the intermediate? Is that what was observed?*

We appreciate your thoughtful discussion of the Remane curve and have updated this section to include our observation of the number of taxa correlated with salinity including intermediate salinity (Rho between -0.25-0.25) and marine (Rho > 0.25). We are limited in our ability to assess freshwater taxa richness as we had a small number (2 of 47 samples) of sites that would classify as freshwater (<0.5 salinity). Moreover, we have updated our Introduction and Discussion to include your suggested comments:

Introduction

Lines 87-102: “Estuarine environments are highly diverse, interconnected ecosystems that are exposed to strong natural fluctuations in salinity and nutrient availability (1–3). Under future climate scenarios, estuaries are expected to have increased erosion and intrusion of saltwater from sea level rise, degraded water quality from runoff and pollution from severe storms, and increased temperatures (4–7), all of which could fundamentally alter microbial community composition and thus the processing of nutrients and food web dynamics (8). Fluctuations in salinity from sea-level rise are especially likely to change microbial community structure and metabolic capacity (8–11). This reflects the fact that organisms require unique metabolic and cellular features in marine and freshwater environments (6, 8, 12–15). These important differences are hypothesized to lead to low species diversity at intermediate salinities and high species diversity in freshwater and marine environments, as first visualized in the Remane curve (16). Although the Remane curve was not originally formulated to describe microorganisms, the pattern was found to apply to phytoplankton and other microbes (17). The relationships described in the curve anticipated the now-supported observation of infrequent transitions between marine and freshwater species (5, 11, 14, 85) and the importance of salinity in structuring microbial communities (4–7, 26).”

Discussion

Lines 322-329: “The fluctuating salinity gradients within estuaries lead to transitory freshwater and marine lineages with fewer well-adapted brackish community members (69), generalized in the Remane curve of species richness across the salinity spectrum (16, 17, 72). Within the nGoM estuaries, a higher number of taxa were strongly correlated to marine environments than intermediate salinity (**Fig. S5**). However, only two of the forty-seven sites sampled represented freshwater (salinity < 0.5; ARD and Swamp), making it difficult to assess the fit of our data to the full Remane curve (**Table S1**: Sampling sites).”

Line 305: FWC is defined here, but used earlier without definition.

Thank you. We have now modified our introduction to introduce the sites prior to the results:

Lines 140-146: “Sites sampled annually for three years included Lake Borgne (LKB; Pontchartrain watershed), Bay Pomme d'Or (JLB; Barataria watershed), Terrebonne Bay (Tbon;

Terrebonne watershed), Atchafalaya River Delta (ARD; Vermilion-Teche/Atchafalaya watershed), Freshwater City (FWC; Mermentau watershed), and the Calcasieu Jetties (CJ; Calcasieu watershed); while sites Sabine Wetlands (Sabine; Calcasieu watershed) and Bay Batiste (Bbat; Barataria watershed) were sampled bi-monthly in 2015; and the inland Lake Martin (Swamp; Vermilion-Teche watershed) was sampled once in 2014.”

Line 312–314: this sentence reads as fragments (comma splice or missed oxford comma?) - suggested to revise this sentence

We have revised this sentence:

Lines 341-344: “Changes in these energetics and metabolisms could fundamentally alter carbon cycling and other nutrient availability. Alterations to these processes can lead to differences in carbon transformation or functional capacity and cascading effects in food web dynamics and nutrient cycling across the ecosystem.”

Line 316 - I didn't quite catch the connection between energetics and management - the implications of changing energetics should be laid out more explicitly

We have revised this section:

Lines 342-346: “Alterations to these processes can lead to differences in carbon transformation or functional capacity and cascading effects in food web dynamics and nutrient cycling across the ecosystem. Therefore, considering the variability in energetics alongside nutrient availability over time and space is critical for effective future conservation and management strategies under future climate and sea-level rise scenarios.”

Line 322: typo, ASV^{is} → ASV^s

Thank you. We have corrected ASV^{is} to ASV^s.

Line 345: maximum abundance → maximum relative abundance

We have corrected all references to abundance to relative abundance.

Line 368–371: awkward phrasing, suggested revision

We have revised this section to help clarify our hypothesis on SAR324's presence in brackish environments:

Lines 396-400: “Analysis of cross-biome transitions found that genome plasticity may be a hallmark feature of brackish-adapted organisms (6). Indeed, members of the SAR324 clade have diverse metabolic capabilities, allowing them to exploit various energy sources and substrates (6, 85), which may have facilitated their ecological and evolutionary transition to brackish environments.”

Line 358–375: the paragraph about SAR324 feels out of order, the previous paragraph describes other potentially brackish-adapted clades and concludes with a disclaimer describing the shortcomings of amplicon sequencing. The SAR324 information could be incorporated into the previous paragraph, or the disclaimer could be moved down.

We have moved the disclaimer from above to alleviate the confusion and disconnection between the two paragraphs. The SAR324 paragraph now ends:

Lines 400-404: “However, given our limited taxonomic and metabolic clarity with amplicon sequencing, further genomic and physiological studies are needed to place these taxa within the associated clades and test their salinity preferences. Future studies should incorporate more estuarine samples to help further delineate the global brackish microbiome and investigate the functional diversity and processes that led to its differentiation.”

377 - Euryhaline is defined here, but is already used above. The definition could be in the introduction or at first use.

Thank you. We have moved this definition to the first reference of euryhaline in the Introduction (lines 149-153).

Line 393: “blasted to” is used here to indicate high nucleotide % identity in BLASTn searches. Generally, blast as a verb refers to doing a BLAST search, not necessarily matching. I recommend rephrasing these results with clearer language.

Thank you. We can see how the term could be confusing, and have revised this section to say “shared high sequence identity” (line 423).

Line 396: Kapabacteriales was italicized in the previous line.

Thank you. We italicized “*Kapabacteriales*”.

Line 408-411: run on

We have corrected this:

Lines 311-315: “Furthermore, the interconnected coastal and estuarine ecosystems are under increasing pressure due to land loss and sea level rise (4, 34). The samples we collected across the nGoM Louisiana coastline represent a diverse collection of environments with notable differences in salinity and other environmental regimes compared to the more stable open ocean.”

411: *data contributes* → *data contribute*

We have changed our data to “our research” (line 315).

Line 417: states that future research should incorporate time series data. There is a temporal element to the data collected here (3 years), but this is not discussed in the results or discussion sections?

Please see our response to Major Issue 2 above. Our sites were sampled across years and, in some cases, during different months. However, most sites sampled did not have a seasonal component to them because they were sampled only once during the year. This limited our ability to discuss temporal correlations and we have highlighted the need for a better temporally designed study to capture seasonal and tidal differences.

To help avoid implying that we could have done a seasonal analysis, we have removed an instance where we attempted to link our data to seasonality of Lines 201-202 (original manuscript): “ASV47 may be further selected for by season as relative abundances were > 0.1% during the late spring and summer months (Fig. 5B; Table S1: *Synechococcus* RA).”

471: *Silva V132* → *SILVA v132*

Thank you. We have corrected this in the manuscript.

477: *PhyloSeq program* → *PhyloSeq package*

Thank you. We have rephrased this to say, “using the PhyloSeq package” (line 512).

510: what are iTag sequences? Are these the raw reads or the ASVs? May be better to avoid new jargon?

Thank you. We have replaced “iTag sequences” with “raw read sequences” (lines 546-551).

Table 1: It is unclear in the table what published/unpublished is or why it is included. The sampling dates show that the samples were not collected in the same month each year—do you expect seasonality? Were they collected at the same time of day / same tide? How would incoming or outgoing tides impact your results?

We address the published versus unpublished column in the results:

Lines 160-163: “Six sites were sampled once a year for three years as part of our nGoM cultivation campaign to compare our isolates to the natural communities, but a comprehensive ecological analysis of these samples was not previously completed. (41, 42) (**Table 1**).”

We included a "published or unpublished" column in Table 1 to transparently indicate which samples had associated data previously reported elsewhere, as a subset of these samples were originally published in a cultivation-focused study. However, the comprehensive microbial community analyses presented in this current manuscript represent novel work that has not been published previously.

We have updated Table 1 columns to include “**Free-living fraction sequences**” and “**Particle-associated fraction sequences**” to help clarify what was published and unpublished.

Lastly, we originally hoped to investigate seasonality using sampling time as a variable, but the inconsistent sampling frequencies across sites precluded this analysis. Tide and hydrological setting were not considered in the interpretation due to lack of detailed hydrological data and temporal controls. Please see our previous comment under Major Issue 2 for further explanation regarding why hydrology and tides were not addressed in this study. We hope to include this approach for future studies.

Figure 1A: Map needs more labeling and a scale. An inset showing where this map sits in a larger map would be helpful. It would also be extremely helpful if the map included rivers with the river names labeled. Also, where there are three points for each site, the shape could reflect the year or years could be labeled. Include full names of sampling locations in the legend.

Thank you for the recommendations. We have modified Figure 1A to enhance its informativeness. The size of the points now represents salinity, while the color indicates different salinity groups. Additionally, we have improved the map by adding a scale bar, a north arrow, waterways, and the outlines of the 10 coastal basins to provide better context for the sample locations. We also included a map insert of Louisiana to help provide context to where the samples are from. We have updated the figure caption (see below) to include full site names and basin location. These changes aim to make the figure clearer and more useful for interpreting the data.

Lines 618-638: “**Figure 1.** A) Locations of the nine sampling sites along the coast of the northern Gulf of Mexico. The shape of the point is the year the sample was collected. The color indicates the broad site salinity classification of fresh (Blue, < 0.5 salinity), low brackish (Orange, < 12 salinity), and high brackish (Red, > 12 salinity). The size of the shape corresponds to the measured salinity. The dotted line outlines the 10 coastal basins. The light blue lines are waterways. The insert is a map of Louisiana highlighting the targeted location of the sampling sites. The map was made with the R packages `rnatrualearth` and `ggplot2`. Shapefiles were obtained from U.S. Geological Survey, National Wetlands Research Center. The sites sampled were Lake Borgne (LKB, Shell Beach, LA) from the Pontchartrain watershed, Bay Pomme d’Or (JLB, Buras, LA) from the Barataria watershed, Terrebonne Bay (TBON, Cocodrie, LA) from the Terrebonne watershed, Atchafalaya River Delta (ARD, Franklin, LA) from the Vermilion-Teche/Atchafalaya watershed, Freshwater City (FWC, Kaplan, LA) from Mermentau watershed, Calcasieu Jetties (CJ, Cameron, LA) from the Calcasieu Watershed, Sabine Wetlands (Sabine, Cameron, LA) from the Calcasieu watershed, Bay Batiste (BBAT, Port Sulphur, LA) from the Barataria watershed, and Lake Martin (Swamp, Breaux Bridge, LA) from the Vermilion-Teche watershed (**Table 1**). B) Two-dimensional principal coordinates analyses plot of normalized water characteristic variables measured at each site. Eigenvectors are scaled to strength. The percent variation each principal coordinate explains is indicated in parentheses adjacent to the component axis. The color indicates the broad site salinity classification of fresh (Blue, < 0.5 salinity), low brackish (Orange, < 12 salinity), and high brackish (Red, > 12 salinity) at the sampling site”

Figures 2 & 4, high salinity is mislabeled as orange in the legends

Thank you. We have corrected the label to say “Red”.

Figures 3 & 5 colors are very difficult to distinguish (especially the purples)

We have modified the colors to differentiate the purple colors used in Figure 3 and Figure 6 . We appreciate your help with improving our figures.

Figure 4: Differential abundance testing - is this pairwise? What package did you use? DESeq2?

Bacterial relative abundances that varied between salinity groups were identified with the Kruskal-Wallis test for all ASVs, followed by pairwise Wilcoxon comparisons with Benjamini-Hochberg correction.

You can find an example of this workflow here: https://riffomonas.org/code_club/2021-06-08-dynamic-stars-and-bars. This work also is cited in the manuscript:

“106. Schloss Patrick D. 2023. The Riffomonas YouTube Channel: An Educational Resource To Foster Reproducible Research Practices. Microbiology Resource Announcements 12:e01310–22.”

As well as our code here:

https://github.com/theaquaticmicrobiologylab/Henson_Thrash_CoastalnGoM

Polygon → *hexagon* (I believe all multisided shapes are polygons, including stars and rectangles)

Thank you. We have amended this figure as recommend by another reviewer. We have updated Figure 4 description to include a graphical legend to improve clarity. Additionally, we have changed our symbols to bi-colored circles and edited our legend to say High vs. Low Brackish; High Brackish vs. Fresh; Low Brackish vs. Fresh s so we’re using the same categories names of salinity.

Figure 5: Revise “across with salinity”. What nonlinear regression method did you use?

We have revised the Figure 5 legend to say:

Lines 662-665: “**Figure 5.** 0.2-2.7 μm fraction ASV relative abundance within key taxonomic clades, A) SAR11 B) *Synechococcus* along the salinity spectrum sampled. Nonlinear regression lines were generated using *geom_smooth* and the method *loess* within the *ggplot* function as a visual aid for relative abundance trends. ASVs are ordered by clade (SAR11) or subcluster (*Synechococcus*) and then in numerical order.”

We have updated the figure to include how nonlinear regressions were made. Specifically, nonlinear regressions were added using *geom_smooth* and the method *loess* (locally estimated scatterplot smoothing) within the *ggplot* function.

Reviewer #2 (Comments for the Author):

Summary

This work represents a survey of the microbial communities of the Northern Gulf of Mexico (nGOM) along the Louisiana (LA) coastline, a region that has been fairly overlooked especially in microbial ecology (though, coastal estuaries in general are also understudied). This study sampled eight coastal sites and one inland freshwater site and determined the microbial communities by sequencing two sized fractions using Illumina 16S amplicon sequencing with 515F-806R primers. The major finding of this work is that microbes of the nGOM on the LA coastline follow a distribution driven mainly by salinity as predicated by the Reman curve, and that particle size and thus turbidity appear to play an important role. The other finding is that

there is a common, core brackish microbiome between many global estuaries. This is significant as most recent studies of the nGOM focused on oil-spill recovery and other stressors, and thus this dataset adds to our understanding of the “normal” microbiome of nGOM. Overall this work is well-written and easy to follow, and moves the field forward in our understanding of these systems. However, there are a few areas of the paper that could be strengthened or clarified as outlined below:

Major Comments

1. A few grammatical and spelling inconsistencies throughout that should be corrected (I have outlined a few of these below in the line-by-line comments).

Thank you for your support. We have corrected the errors and inconsistencies throughout the manuscript.

2. Some of the terminology is a bit unclear, specifically “high vs low brackish” and size fractions:

a. What is the justification for the cutoff of this definition at salinity 12? Should add a line just mentioning why that was chosen. The terms themselves are a tad arbitrary; unfortunately the “oligohaline, mesohaline” definitions are unhelpful for your study so I don’t have a better suggestion – I think explaining why 12 was chosen as the divisor is sufficient.

Indeed, traditional definitions of salinity are difficult to apply to our study. Moreover, general classification makes it difficult to determine differences in salinity since brackish is defined as salinities between 0.5-30. Using our NMDS ordination to help define cut-off values, we found a strong separation along the vertical 0 axes of the NMDS ordination of beta diversity with salinities above and below 12. We therefore separated our brackish sites into low brackish (0.5-12 salinity) and high brackish (> 12 salinity). We have added text to help clarify this:

Lines 182-191: “Non-metric multidimensional scaling (NMDS) of the 47 communities showed three distinct groupings based on salinity: fresh (< 0.5 salinity), low brackish (0.5-12), and high brackish (> 12 salinity) (Fig. 2**; NMDS stress 0.135; ANOSIM R=0.703, p=0.001). Moreover, salinity ($R^2=0.807$, p=0.001) and silicic acid ($R^2=0.563$, p=0.001) were the two strongest environmental variables correlated to the NMDS ordination of the combined fraction analysis (**Fig. 2**; **Table S1**: envfit). While “brackish” is usually defined as salinities between 0.5 and 30, we observed a significant separation at the vertical axis of our NMDS ordination between sites with salinities above and below 12, and thus used this to separate high from low brackish categories. Filter fraction difference (free-living vs. particle-associated) was significant but had low explanatory power (ANOSIM R=0.244, p=0.001).”**

b. Is there a reason for choosing to refer to 0.2-2.7 μm and $>2.7\mu\text{m}$ constantly instead of just calling them free-living (FL) vs particle-associated (PA)? I know there are many caveats with these terms, but I think it makes it harder to process the numbers constantly and many other groups use similar cutoffs and define/justify that labeling. Perhaps this was taken out in an earlier revision, as “particle-associated fraction” (without the numbers) is mentioned at L244.

As suggested, we have removed the numerical references and replaced them with particle-associated and free-living fractions. First referenced here:

Lines 456-458: “We refer to fractions collected on the 2.7 μm and 0.22 filters as particle-associated ($> 2.7 \mu\text{m}$) and free-living (0.22-2.7 μm) fractions, respectively.”

3. Seasonality and tidal influence in sampling is mentioned in some parts of the paper, but given the low number of samples this has to be carefully discussed (especially since very few were sampled during the same date/month or season) – hard to give much of a temporal angle. Many of the sites have single-season samples, and it does not appear that tides were specifically included in the environmental analysis – not that you have to, but curious if it might have an impact/is worth mentioning if you were consistent in which tidal period you sampled. Were any other environmental variables like tides sampled or considered (e.g., river flow, precipitation and other meteorological values)

For this analysis, we did not collect river flow or precipitation. We only collected data on parameters associated with the sampled water (e.g., salinity, temperature). We agree with the reviewer on the importance of considering these other variables for future studies. We have limited our discussion of the seasonality of our samples and removed the reference to *Synechococcus* and season from our results (see comment below). Also, we include our response to a similar comment from Reviewer 1 here:

When we first began this project, we hoped we could use sampling time as one of our variables and maybe suss out seasonality. However, the sampling frequencies across sites were too different to use temporal information. Tide and hydrological setting were not considered in the interpretation. Without detailed hydrological information about lag times between rain pulses and delivery to our sampling sites, as well as temporal controls before and after such events, we cannot make any assertions about the effect of hydrology on microbial communities. The study was not designed for such analyses. Although not addressed directly, tidal effects at these locations should be minimal because of the mostly diurnal nature of the tides and the low tidal exchange in the region. For example, the tidal range during all of March of this year near our Atchafalaya sampling site spanned only 2 feet (-0.5 – 1.5). Regardless, we did not plan the sampling around tides, and therefore cannot comment on the effect of tides in the study.

4. Differences between size fractions – in the results, you mention strongly that the diversity increase in particle-associated communities is due to filters clogging. Though this is a good acknowledgement of the potential impact; however, I think this actually undermines your study, where you have enough data to test whether this might have happened. Especially since your filter volumes were not exceptionally high (150 mL), if that was consistent throughout and you can note when/how often they clogged (if different among sites), that at least provides consistency across the study. Additionally, it could be worth mentioning how much overlap there was between the size fraction communities (as a %, etc.) – are there known free-living taxa you found in >2.7µm? Is this so prevalent you can confirm clogging, or is it actually not that large of a proportion relatively?

We agree this is a complicated but important topic. We unfortunately did not record specifically which specific samples were clogged, the information is from informal data (e.g., oral communication). We have amended our results section to add in the known free-living taxa found within our particle-associated fraction and their presence within the top 25 most abundant taxa in the particle-associated fraction:

Lines 234-237: “Within the top twenty-five abundant taxa in the particle-associated fraction (average relative abundance), thirteen were classified as known planktonic organisms, with median relative abundances > 0.1%, max abundances between 0.25-3%, and representing about 11% of the total reads (**Fig. S1; Table S1**: particle-associated RA).”

Regarding clogging, we caution the readers on the interpretation of our data, especially when examining microbial diversity:

Lines 249-252: “Although we did not investigate the impact of filtered volume on which taxa were observed in the 2.7 µm filters, our results highlight that microbial ecologists should consider, and experimentally validate, how sediment load, in addition to volume, impacts size fractionated communities when working in coastal and estuary environments with high turbidity.”

Lastly, we addressed some previous research on this topic in our manuscript:

Example lines 242-244: “ Indeed, the amount of volume filtered can act as a secondary filter, trapping planktonic cells on the prefilter (> 2.7 µm fraction) and biasing downstream analyses (55).”

Minor Comments

1. Could improve the title itself to reflect estuarine waters (sounds more marine than it is, which might be misleading). Potentially just adding “estuarine” could work?

We have provided a new title to highlight the estuarine aspect of this work:

“Microbial ecology of northern Gulf of Mexico estuarine waters”

2. *Importance can be stronger (see below L82)*

We have modified our Importance statement (see below for addressing the specific comment).

3. *It seems that a prior version of this might have been written with the Materials and Methods coming before the Results – much of the results need a little introductory context since the methods come so much later (e.g., missing a mention of which specific environmental measurements and sites were sampled prior to R and p values)*

Unfortunately, mSystems does not allow for methods to be moved earlier. Therefore, we have included site descriptions prior to their discussion in the results:

Lines 140-146: “ Sites sampled annually for three years included Lake Borgne (LKB; Pontchartrain watershed), Bay Pomme d'Or (JLB; Barataria watershed), Terrebonne Bay (Tbon; Terrebonne watershed), Atchafalaya River Delta (ARD; Vermilion-Teche/Atchafalaya watershed), Freshwater City (FWC; Mermentau watershed), and the Calcasieu Jetties (CJ; Calcasieu watershed); while sites Sabine Wetlands (Sabine; Calcasieu watershed) and Bay Batiste (Bbat; Barataria watershed) were sampled bi-monthly in 2015; and the inland Lake Martin (Swamp; Vermilion-Teche watershed) was sampled once in 2014.”

4. *Description of the sites could be added at the end of introduction (see below) – for clarification, Are any of these sites interconnected? Does water flow from the freshwater sites through the low brackish and into the high brackish, or are sites representative of unique watershed may require their own analysis or discussion?*

As suggested, we have included a brief description of the sites in the introduction:

Lines 140-146: “ Sites sampled annually for three years included Lake Borgne (LKB; Pontchartrain watershed), Bay Pomme d'Or (JLB; Barataria watershed), Terrebonne Bay (Tbon; Terrebonne watershed), Atchafalaya River Delta (ARD; Vermilion-Teche/Atchafalaya watershed), Freshwater City (FWC; Mermentau watershed), and the Calcasieu Jetties (CJ; Calcasieu watershed); while sites Sabine Wetlands (Sabine; Calcasieu watershed) and Bay Batiste (Bbat; Barataria watershed) were sampled bi-monthly in 2015; and the inland Lake Martin (Swamp; Vermilion-Teche watershed) was sampled once in 2014.”

We have also modified our methods to include more information:

Lines 468-487: “Sites were sampled once a year for three years, except for Terrebonne Bay, which was collected twice. The sites sampled were Lake Borgne (LKB, Shell Beach, LA) from the Pontchartrain watershed, Bay Pomme d’Or (JLB, Buras, LA) from the Barataria watershed, Terrebonne Bay (TBON, Cocodrie, LA) from the Terrebonne watershed, Atchafalaya River Delta (ARD, Franklin, LA) from the Vermilion-Teche/Atchafaya watershed, Freshwater City (FWC, Kaplan, LA) from Mermentau watershed, and Calcasieu Jetties (CJ, Cameron, LA) from the Calcasieu Watershed (**Table 1**). The sample collection was done previously as part of the three-year cultivation campaign between September 2014 and February 2017 (43).

Estuarine nGoM sites

Two sites were sampled once roughly every two months for five months between July 2015 and November 2015. The sites sampled were Sabine Wetlands (Sabine, Cameron, LA) from the Calcasieu watershed and Bay Batiste (BBAT, Port Sulphur, LA) from the Barataria watershed (**Table 1**). Water was collected from the surface water (< 1 m) and filtered immediately on site as described above.

Lake Martin

Lake Martin (Swamp, Breaux Bridge, LA) from the Vermilion-Teche watershed was sampled as an inland lake representative in November 2014 (**Table 1**). Water was collected from the surface water (< 1 m) and filtered immediately on site as described above.”

The sites within each watershed (Sabine and CJ in Calcasieu, Bbat and JLB in Barataria) are interconnected. However, due to differences in sampling frequencies and timing across all sites, we did not directly compare intra- and inter-watersheds for this study, though future investigation of this would be interesting.

5. A basic description of your sequencing results should be included around L161 – they are mentioned in the results, but since those appear later it would be nice to just get a few lines with “We recovered # sequences and # ASVs from different sites” in whatever level of detail you’d like. Just seems to miss a link between your PCA results and going right to NMDS using sequencing data.

Thank you for this comment. We have added the following information to our manuscript between the PCA and NMDS descriptions:

Lines 178-181: “After quality filtering and rarifying, we recovered 7,341 ASVs from the nine sites. ASVs were predominately classified into four bacterial phyla (82% of all reads) – Proteobacteria (36%), Actinobacteriota (20%), Bacteroidota (15%), and Cyanobacteria (11%) –

and three archaeal phyla – Thermoplasmata (0.3%), Crenarchaeota (0.3%), and Nanoarchaeota (0.2%).”

6. *For Fig. 2, is ord fit appropriate for NMDS? Often I have seen this just as a CCA analysis with arrows; okay if so, but would be good to add a line of justification.*

Using *envfit* on an NMDS plot of microbial OTUs enhances the interpretability of the data by revealing the environmental factors that shape microbial community structure and facilitating hypothesis testing to determine the significance of these relationships. *envfit* is designed to fit environmental vectors or factors onto an ordination. NMDS is a distance-based matrix ordination, so it is appropriate for *envfit*. We have provided an abbreviated list of studies below and slightly modified our methods to provide an expanded explanation:

Lines 521-523: “*envfit* was used to interpret which environmental parameters were significantly contributing to the NMDS ordination by fitting vectors of significant variables onto the beta diversity-based distance matrix (105).”

Here are a few papers that used similar methods:

<https://doi.org/10.1002/Ino.11130>

<https://doi.org/10.3389/fmicb.2019.02628>

<https://doi.org/10.1093/femsec/fiz033>

<https://doi.org/10.1038/ismej.2015.226>

7. *Within the discussion, comparison with any other nGoM coastal sequencing is lacking. I know there has been very little data in this realm, but it could be worth mentioning any similar groups or novel groups recovered in your study vs others.*

We have updated our sections to more clearly state the findings from these early Gulf of Mexico, Louisiana continental shelf studies.

In our Introduction, our overview of the studies now says:

Lines 130-134: “Samples from pre-Deepwater Horizon oil spill microbial communities along the continental Louisiana shelf were dominated by Alphaproteobacteria and Bacteroidota, specifically the SAR11 clade (30). In contrast, deeper samples found an increasing abundance of Archaea, specifically *Crenarchaeota* (a.k.a. Thaumarchaeota, Nitrososphaerota) (42).”

In our Results:

Lines 199-203: “At higher salinities (> 12), typical marine clades such as SAR11 subclade I (Alphaproteobacteria) and *Candidatus Actinomarina* (OM1 clade, Actinobacteria) were more abundant than in samples of lower salinity (**Fig. 4A**), supporting previous observations of their high relative abundance in samples from the Louisiana continental shelf with high salinity (30).”

In our Discussion:

Lines 390-391 “In the Gulf of Mexico, SAR324 was dispersed across the Louisiana continental shelf, albeit at low relative abundances (30, 87).”

8. *Why not Domain instead of Kingdom? Perhaps this is a relic from the new Silva Output?*

The output from the SILVA taxonomy is "Kingdom," "Phylum," "Class," "Order," "Family," and "Genus." We have kept this labeling scheme to prevent confusion.

9. *As written, the word Archaea is never mentioned in the text, yet MGII is presented (L263) without defining. Since you recovered more than MGII Eury, could you add a line briefly describing which archaeal groups you recovered somewhere, even if they did not show a seasonal context as well? In line with under-sampling of these coastal estuaries in microbiology, Archaea are often ignored and you do have them!*

We have updated our results of the general findings to include the archaeal phyla found:

Lines 178-181: “After quality filtering and rarifying, we recovered 7,341 ASVs from the nine sites. ASVs were predominately classified into four bacterial phyla (82% of all reads) – Proteobacteria (36%), Actinobacteriota (20%), Bacteroidota (15%), and Cyanobacteria (11%) – and three archaeal phyla – Thermoplasmatota (0.3%), Crenarchaeota (0.3%), and Nanoarchaeota (0.2%).”

10. *The claim for a RS62 Beta bloom (ASV25) at FWC is potentially a stretch given only three samples, though context with other studies may support. Might be worth defining/distinguishing. Are any of the other sites similar enough to the Vermilion River Estuary for comparisons? Could this be an artifact of tides (e.g., perhaps RS62 is present at high tide and comes from the ocean)?*

We agree with the reviewer that the RS62 bloom hypothesis is circumstantial and have rephrased these results to say:

Lines 277-281: “It is difficult to distinguish if the abundance of RS62 was correlated to an unmeasured phytoplankton bloom or if it was a remnant population flushed in during tidal exchange. RS62 taxa occurred in high relative abundances in the planktonic fractions in the Pearl

River estuary system (63) but had strong associations with phytoplankton blooms elsewhere (67).”

Specific Line Comments

Abstract & Importance

• L68: *autochthonous and allochthonous should be defined in some capacity (even a simple “native versus not” would help, esp. at L273) – I don’t believe allochthonous is mentioned outside of the abstract.*

We have rephrased this to remove these two terms and replaced them with “native” and “transitory”:

Lines 67-70: “The data presented here expand the geographic coverage of microbial ecology in estuarine communities, help delineate the native and transitory members of these environments and provide critical aquatic microbiological baseline data for coastal and estuarine sites in the nGoM.”

• L82-84: *This is the most impactful part of the importance. While I (microbial ecologist) totally agree with the importance otherwise, a broader group might not “get” why they should care about characterizing estuarine microbes – why does this diversity help economic value? (Mostly, I think you could rephrase a bit and make this more impactful to your audience!)*

We appreciate your thoughtful comments. We have rewritten our importance statement with your comments in mind:

Lines 73-85: “Estuarine and coastal waters are diverse ecosystems influenced by tidal fluxes, interconnected wetlands, and river outflows, which are of high economic and ecological importance. Microorganisms play a pivotal role in estuaries as “first responders” and ecosystem architects, yet despite their ecological importance, they remain underrepresented in microbial studies compared to open ocean environments. This leads to substantial knowledge gaps that are important for understanding global biogeochemical cycling and making decisions about conservation and management strategies in these environments. Our study makes key contributions to the microbial ecology of estuarine and coastal habitats in the northern Gulf of Mexico. Our microbial community data supports the concept of a globally distributed, core brackish microbiome and emphasizes previously underrecognized brackish-water taxa. Given the projected worsening of land loss, oil spills, and natural disasters in this region, our results will

serve as important baseline data for researchers investigating the microbial communities found across estuaries.”

Introduction:

• *L92: “circumventing the marine-freshwater threshold” is not quite clear – I assume you mean “crossing the salinity divide” essentially, but as written needs a little more context.*

We have rephrased this to clarify what was being discussed:

Lines 92-101: “Fluctuations in salinity from sea-level rise are especially likely to change microbial community structure and metabolic capacity (8–11). This reflects the fact that organisms require unique metabolic and cellular features in marine and freshwater environments (6, 8, 12–15). These important differences are hypothesized to lead to low species diversity at intermediate salinities and high species diversity in freshwater and marine environments, as first visualized in the Remane curve (16). Although the Remane curve was not originally formulated to describe microorganisms, the pattern was found to apply to phytoplankton and other microbes (17). The relationships described in the curve anticipated the now-supported observation of infrequent transitions between marine and freshwater species (9, 15, 17, 18) and the importance of salinity in structuring microbial communities (8–11, 19).”

• *L125: Not all of these were sampled during the oil spill; two contain pre-spill samples only (33 & 35), so perhaps need some rephrasing.*

We have rephrased this sentence to clarify that we were emphasizing the lack of general microbial ecology samples from the nGoM estuaries. We have also included a description of what previous studies from the shelf have found:

Lines 125-134: “microbiological research in this region has mostly focused on the communities associated with oil spills and eutrophication, and there is much less data on the general microbial ecology from the estuary ecosystems across the nGoM (36–41). Samples from pre-Deepwater Horizon oil spill microbial communities along the continental Louisiana shelf were dominated by Alphaproteobacteria and Bacteroidota, specifically the SAR11 clade (30). In contrast, deeper samples found an increasing abundance of Archaea, specifically *Crenarchaeota* (a.k.a. Thaumarchaeota, Nitrososphaerota) (42). However, limited investigation of the microbial ecology from nGoM estuaries has restricted our ability to infer how bacterioplankton in the nGoM naturally fluctuate over time in these diverse habitats.”

• L128: *Would be helpful to mention site names and define them at this point of the introduction; otherwise they are not mentioned except for a Table (methods coming later make it hard to follow results without some intro).*

Unfortunately, mSystems does not allow us to move the methods up. As suggested, we have included a brief description of the sites in the introduction:

Lines 130-146: “ Sites sampled annually for three years included Lake Borgne (LKB; Pontchartrain watershed), Bay Pomme d'Or (JLB; Barataria watershed), Terrebonne Bay (Tbon; Terrebonne watershed), Atchafalaya River Delta (ARD; Vermilion-Teche/Atchafalaya watershed), Freshwater City (FWC; Mermentau watershed), and the Calcasieu Jetties (CJ; Calcasieu watershed); while sites Sabine Wetlands (Sabine; Calcasieu watershed) and Bay Batiste (Bbat; Barataria watershed) were sampled bi-monthly in 2015; and the inland Lake Martin (Swamp; Vermilion-Teche watershed) was sampled once in 2014.”

Results and Discussion

• L156-159: *The phrasing “above the 0 line” is really confusing as written. I think it would be easier for the reader if you focused on calling the axes as labeled in the figure (PC1 and PC2) and using positive/negative direction instead of above and below (or left/right). For example, L156 would then read: “Sites found along the positive PC2 axis typically had higher salinities”*

Thank you for your comments on improving the clarity of our results. We have corrected these results to better explain where sites fell on the PCA plot:

Lines 171-177: “Principal component analysis of environmental conditions at the nine sites showed a distinct separation along both the PC1 and PC2 axes, which together explained more than half of the variance (**Fig. 1B**). Salinity and NO_2^- were the most important variables separating the sites. Sites along the positive PC2 axis line typically had higher salinities (> 12) and higher NO_2^- and NH_4^+ concentrations. In contrast, sites along the negative PC2 axis were more indicative of high silicic acid (**Fig. 1B**). Sites along the positive PC1 axis typically had higher nitrate and phosphate concentrations (**Fig. 1B**).”

• L185: *Syn is misspelled*

Thank you. We have corrected the spelling of *Synechococcus*.

• L201: This figure does not represent seasonality quite well – is there another spot readers could look to examine seasonal differences in these populations? (though, see above – not many samples have a seasonal survey).

Given the lack of samples across seasons for each site, we have removed this line from the text.

- L218: *Syn* not italicized

Thank you. We have italicized *Synechococcus*.

- L220: *While I agree with you, I feel like the notion of coastal researchers needing to consider sediment load in their study is pretty well understood. Perhaps the real novelty of your study is it is rarely experimentally validated? Though there may be some early work that you could cite here that tested this.*

We have reworded this sentence to reflect better the importance of microbial ecologists' consideration and validation:

Lines 249-252: “Although we did not investigate the impact of filtered volume on which taxa were observed in the 2.7 μm filters, our results highlight that microbial ecologists should consider, and experimentally validate, how sediment load, in addition to volume, impacts size fractionated communities when working in coastal and estuary environments with high turbidity.”

- L230-231: *Here, “silicic acid” is used whereas elsewhere in the paper the chemical formula is mentioned. Should be consistent.*

We have updated the first silicic acid reference in manuscript to “silicic acid $[\text{Si}(\text{OH})_4]$ ”.

- L254: *Euryhaline should be defined (perhaps trade from L377-378)*

We have moved this definition from L377 to the first reference of euryhaline in the Introduction (lines 149-153).

Discussion

- L289-292: *This is close to copy/pasted to the Importance, and could be rephrased.*

We have improved this section by rephrasing its importance:

Lines 315-320: “Our research contributes valuable insights into the microbial ecology of estuarine and coastal ecosystems, emphasizing the need for further studies to elucidate population-level diversity, functional roles, and temporal variations in these dynamic environments. These data will serve as both a baseline and an important resource for future

researchers investigating the microbial communities found across the coastal nGoM and estuaries globally.”

• L309: *the link of shifting carbon metabolisms feels like it needs a bit more support or justification to align with your study and results.*

We have altered this section to improve clarity and justification:

Lines 341-346: “Changes in these energetics and metabolisms could fundamentally alter carbon cycling and other nutrient availability. Alterations to these processes can lead to differences in carbon transformation or functional capacity and cascading effects in food web dynamics and nutrient cycling across the ecosystem. Therefore, considering the variability in energetics alongside nutrient availability over time and space is critical for effective future conservation and management strategies under future climate and sea-level rise scenarios.”

• L369: *CBB cycle is misspelled (Calvin not Carbon, Bassham)*

Thank you. We have corrected Carbon to Calvin.

Materials and Methods

• L450: *It is Breaux Bridge (also correct on tables)*

Thank you. We have corrected this typo.

• L494: *Two references that are not numbered here (spelled out instead)*

We specifically mention the papers associated with the sequence data for each clade. While we have left in the spelled-out citations, we have added the numbered citations to the end of the sentence.

• L499: *principal (not principle) component (not components)*

We have corrected this error in the manuscript. Thank you.

Figures and Tables

• Table 1: *Could you split this for individual dates? The ranges I understand are more summative, but given the huge variability in each site, I think it would be much more valuable for readers to have each date separate and then group the rows by site and publication status.*

We agree that the variability found at the sites can range drastically. We provided the ranges to prevent Table 1 from being very large, while Table S1 ‘Sampling Sites’ tab contains the larger list. To help point the readers to the larger list, we have amended the methods section to include the statement:

Lines 463-465: “All site locations (latitude and longitude), water chemistry, water conditions, and sampling dates can be found in **Table S1** in the ‘Sampling Sites’ tab.”

Moreover, we have amended our Introduction to include a brief introduction to the sites:

Lines 140-146: “ Sites sampled annually for three years included Lake Borgne (LKB; Pontchartrain watershed), Bay Pomme d'Or (JLB; Barataria watershed), Terrebonne Bay (Tbon; Terrebonne watershed), Atchafalaya River Delta (ARD; Vermilion-Teche/Atchafalaya watershed), Freshwater City (FWC; Mermentau watershed), and the Calcasieu Jetties (CJ; Calcasieu watershed); while sites Sabine Wetlands (Sabine; Calcasieu watershed) and Bay Batiste (Bbat; Barataria watershed) were sampled bi-monthly in 2015; and the inland Lake Martin (Swamp; Vermilion-Teche watershed) was sampled once in 2014.”

• *Table S1: wrong sampling date listed for JLB2C; possible to add tidal stage on sampling dates/times?*

We have corrected the wrong sampling date for JLB2C in Table S1.

Thank you for the suggestion to include tidal stage information. Unfortunately, we do not have specific times of samples for all sites. While tides are relevant in coastal ecosystems, the tidal range in this region is relatively small. Moreover, to properly assess tidal impacts, sampling would need to be conducted throughout complete tidal cycles, which was beyond the scope of our experimental design and sampling regime. As mentioned earlier, our study lacked the detailed hydrological data (e.g., lag times between precipitation and site delivery) required to make robust inferences about the effects of hydrology, including tides, on the microbial communities. However, we appreciate the suggestion and agree that considering tidal dynamics would be prudent for related research in this region going forward.

• *Fig. 1 Legend: Components not Coordinates (based on the rest of the paper, L584); add description of colors/categories.*

We have changed components to coordinates.

• *Fig. 2 Legend: mention envfit analysis (L593)*

We have amended this legend to include the use of *envfit*:

Lines 646-647: “Significant environmental variables ($p < 0.05$) determined with *envfit* are plotted as vectors. Arrow lengths have been adjusted based on their strength of correlation (R^2).”

- *Fig. 4: Hard to interpret shapes; at minimum, add graphical legend, but could consider perhaps bi-color circle symbols to reflect your significant comparisons? (e.g., circle that is half-red and half-blue).*

Thank you for the comments on how to improve this graph. We have updated Figure 4 to include a graphical legend to improve clarity. Additionally, as suggested, we have changed our symbols to bi-colored circles. Lastly, we have edited our legend to say High vs. Low Brackish; High Brackish vs. Fresh; Low Brackish vs. Fresh so we’re using the same categories

- *Fig. 5: Are these ordered by similar trendlines? If so, is it possible to explain or color-code specialist vs. generalist, or add to the legend?*

These are ordered by subclade (SAR11) or subcluster (Syn) and then by numerical order. We have updated the figure legend to clarify (lines 661-662):

Lines 662-666: “**Figure 5.** Free-living fraction ASV relative abundance within key taxonomic clades, A) SAR11 B) *Synechococcus* along the salinity spectrum sampled. Nonlinear regression lines were generated using *geom_smooth*, and the method *loess* within the *ggplot* function as a visual aid for relative abundance trends. ASVs are ordered by subclade (SAR11) or subcluster (*Synechococcus*) and then listed in numerical ASV order.”

- *Fig. 6 Legend: explain that arrow is fresh vs marine sites in-text.*

We have added the following text to the Figure 6 legend:

Lines 671-674: “The bi-directional arrow above the dot plots provides an orientation of correlations to potential salinity associations. Negative rho values correspond to potential freshwater associations. Positive rho values correspond to potential marine associations.”

- *Fig. S5: Add fresh vs marine arrow as in Fig. 6*

We have added an arrow above Fig. S5.

References

- #18: *is this a complete reference? Seems to be missing journal/volume/page.*

We have updated this book chapter to be correctly formatted:

Lines 756-761: “Nicholls RJ, Wong PP, Burkett V, Codignotto J, Hay J, McLean R, Ragoonaden S, Woodroffe CD, Abuodha PAO, Arblaster J, Brown B, Forbes D, Hall J, Kovats S, Lowe J, McInnes K, Moser S, Rupp-Armstrong S, Saito Y. 2007. Coastal systems and low-lying areas, p. 315–356. *In* Parry, ML, Canziani, OF, Palutikof, JP, van der Linden, PJ, Hanson, CE (eds.), *Climate Change 2007: impacts, adaptation and vulnerability*. Cambridge University Press.”

• #55: *do they all need a DOI?*

We have now removed all DOIs to be consistent with all references.

Reviewer #3 (Comments for the Author):

In this manuscript, the authors present a robust dataset of high throughput 16S rRNA gene amplicon sequencing survey used to investigate spatiotemporal patterns of the microbiome diversity in the northern Gulf of Mexico (nGoM) waters along the Louisiana coast. Nine sampling sites represented a wide range of salinity from freshwater inland to high-brackish coastal waters. Furthermore, particle-associated and planktonic communities were separated at the time of sampling by serial filtration and were investigated separately. Habitat preference driven primarily by salinity of many known freshwater and marine clades were observed.

Overall, this study provides important microbial community baseline data focused on coastal habitats. Below are some suggestions for improvement.

Specific comments:

What was the overall distribution of bacteria and archaea across samples? A brief description would be helpful, perhaps aided by a supplementary figure showing overall community composition at phylum level.

We have added a brief description of the overall composition into the results to discuss phylum-level diversity:

Lines 178-182: “After quality filtering and rarifying, we recovered 7,341 ASVs from the nine sites. ASVs were predominately classified into four bacterial phyla (82% of all reads) – Proteobacteria (36%), Actinobacteriota (20%), Bacteroidota (15%), and Cyanobacteria (11%) – and three archaeal phyla – Thermoplasmatota (0.3%), Crenarchaeota (0.3%), and Nanoarchaeota (0.2%).”

Moreover, we have Fig. S1, which represents the top 25 ASVs from each fraction as a general overview.

Lines 132 - 137: "Six sites ... sampled only once (Tables 1 and S1)." - move these lines to the first paragraph of the Results section.

We have moved this to the first paragraph of the Results line 250. We have also amended our introduction as recommended by other reviews to introduce the sites prior to the Results:

Lines 140-146: "Sites sampled annually for three years included Lake Borgne (LKB; Pontchartrain watershed), Bay Pomme d'Or (JLB; Barataria watershed), Terrebonne Bay (Tbon; Terrebonne watershed), Atchafalaya River Delta (ARD; Vermilion-Teche/Atchafalaya watershed), Freshwater City (FWC; Mermentau watershed), and the Calcasieu Jetties (CJ; Calcasieu watershed); while sites Sabine Wetlands (Sabine; Calcasieu watershed) and Bay Batiste (Bbat; Barataria watershed) were sampled bi-monthly in 2015; and the inland Lake Martin (Swamp; Vermilion-Teche watershed) was sampled once in 2014."

Fig. 1A: In most figures, three sample groups (based on salinity) were displayed using a defined color pattern. Please consider retaining the same color pattern in Fig. 1A for consistency. Salinity values as a continuous variable can still be shown with varying symbol size.

Thank you for the recommendations. We have modified Figure 1A to enhance its informativeness. The size of the points now represents salinity, while the color indicates different salinity groups. Additionally, we have improved the map by adding a scale bar, a north arrow, Louisiana waterways, and the outlines of the 10 coastal basins to provide better context for the sample locations. These changes aim to make the figure clearer and more useful for interpreting the data.

In Figs 3 and 5 relative abundance was shown as fraction while in Fig. 4 it was shown as percentage. A consistent unit of relative abundance across all figures would be helpful for readers.

Thank you for finding this discrepancy. We have now changed Figure 4 to be a fraction.

Line 425: Incomplete parentheses... "(Becton-Dickinson, USA" Replace "gram-negative phylum" with "phylum of gram-negative bacteria".

We have added a parenthesis to the end of USA. However, we are unclear what the reviewer was referencing by 'Replace "gram-negative phylum" with "phylum of gram-negative bacteria".'

Lines 459 - 461: During PCR amplification, how many replicate reactions per sample were conducted?

PCR amplification and prep were conducted at Argonne National Labs. Three replicate reactions per sample were conducted and pooled. The protocol is contained in the reference Apprill et al. 2015. We have moved the citation to the end of the sentence to indicate where protocol and primer sets are from:

Lines 494-495: “DNA was sequenced targeting the 16S rRNA gene V4 region with the 515F-806R primer set at Argonne National Laboratory using Illumina MiSeq 2 x 250 bp paired-end reads (99, 100).”

Line 464: What was the read distribution across samples?

We have added the sentence “Samples ranged in the total number of sequences from 4,785 to 152,624.” (lines 509-510) to the methods section to address this.

Line 471: Why was the most updated version of SILVA (v138.1) not used for taxonomy assignment?

Good catch. Indeed, v138.1 was used for sequence classification. We have updated this in the manuscript.

Lines 510 - 514: Please replace "iTag sequences" with "raw sequence reads".

We have replaced “iTag sequences” with “raw read sequences” (lines 546-551).

Lines 591 & 602: Please use a distinct color name (other than orange) for the high-brackish group.

We have corrected orange to “Red” for the high-brackish group.

Re: mSystems01318-23R1 (**Microbial ecology of northern Gulf of Mexico estuarine waters**)

Dear Dr. Michael Henson:

Your manuscript has been accepted. I find that the responses to reviewers were thorough and generally addressed their major concerns. As such, I see no need to send the manuscript for additional reviews.

One note, on L433 - a citation error is indicated. I am sure the copy editors will identify this, but you can get a head start on the correction.

I am forwarding the manuscript to the ASM production staff for publication. Your paper will first be checked to make sure all elements meet the technical requirements. ASM staff will contact you if anything needs to be revised before copyediting and production can begin. Otherwise, you will be notified when your proofs are ready to be viewed.

Sincerely,
Ryan Newton
Editor
mSystems